# A single-cell nanocoating of probiotics for enhanced amelioration of antibiotic-associated diarrhea

Jiezhou Pan[1,9], Guidong Gong [1,9], Qin Wang[2], Jiaojiao Shang[1], Yunxiang He [1], Chelsea Catania [3], Dan Birnbaum[4], Yifei Li [1,5], Zhijun Jia[1,5,6], Yaoyao Zhang [1,5✉], Neel S. Joshi [4,7✉] & Junling Guo [1,4,8✉]

The gut microbiota represents a large community of microorganisms that play an important role in immune regulation and maintenance of homeostasis. Living bacteria receive increasing interest as potential therapeutics for gut disorders, because they inhibit the colonization of pathogens and positively regulate the composition of bacteria in gut. However, these treatments are often accompanied by antibiotic administration targeting pathogens. In these cases, the efficacy of therapeutic bacteria is compromised by their susceptibility to antibiotics. Here, we demonstrate that a single-cell coating composed of tannic acids and ferric ions, referred to as 'nanoarmor', can protect bacteria from the action of antibiotics. The nanoarmor protects both Gram-positive and Gram-negative bacteria against six clinically relevant antibiotics. The multiple interactions between the nanoarmor and antibiotic molecules allow the antibiotics to be effectively absorbed onto the nanoarmor. Armored probiotics have shown the ability to colonize inside the gastrointestinal tracts of levofloxacin-treated rats, which significantly reduced antibiotic-associated diarrhea (AAD) resulting from the levofloxacin-treatment and improved some of the pre-inflammatory symptoms caused by AAD. This nanoarmor strategy represents a robust platform to enhance the potency of therapeutic bacteria in the gastrointestinal tracts of patients receiving antibiotics and to avoid the negative effects of antibiotics in the gastrointestinal tract.

[1] BMI Center for Biomass Materials and Nanointerfaces, College of Biomass Science and Engineering, West China Second University Hospital, Sichuan University, Chengdu, Sichuan 610065, China. [2] School of Pharmacy, Southwest Minzu University, Chengdu, Sichuan 610065, China. [3] Department of Mechanical Engineering, Massachusetts Institute of Technology, Cambridge, MA 02139, USA. [4] Wyss Institute for Biologically Inspired Engineering, John A. Paulson School of Engineering and Applied Sciences, Harvard University, Cambridge, MA 02138, USA. [5] Key Laboratory of Birth Defects and Related Women and Children of Ministry of Education, Department of Pediatrics, The Reproductive Medical Center, Department of Obstetrics and Gynecology, West China Second University Hospital, Sichuan University, Chengdu, Sichuan 610041, China. [6] Department of Biopharmaceutics, West China School of Pharmacy, Sichuan University, Chengdu, Sichuan 610041, China. [7] Department of Chemistry and Chemical Biology, Northeastern University, Boston, MA 02115, USA. [8] State Key Laboratory of Polymer Materials Engineering, Sichuan University, Chengdu, Sichuan 610065, China. [9] These authors contributed equally: Jiezhou Pan, Guidong Gong. ✉email: zhangyaoyao@hsc.pku.edu.cn; ne.joshi@northeastern.edu; junling.guo@scu.edu.cn

The human gut microbiota plays a critical role in maintaining healthy gastrointestinal (GI) functions and other physiological processes[1–6], which leads to increased interest in the use of orally administered microbes as therapeutics and diagnostics[7–9]. The basic idea behind these strategies leverages the ability of some microbes, such as those labeled 'probiotics', to survive oral and gastric transit and flourish in the intestinal tract, thereby exerting a beneficial biological effect on the host[10–12], including the protection against enteric pathogens, immune modulation, changes in nutrient absorption, and neurological effects[13–15].

Antibiotics are among the most prescribed medications in the world, and their usage continues to increase dramatically[16,17]. However, they are nonspecific in their killing action, leading to a drastic depletion of beneficial gut microbiota simultaneously with the elimination of pathogens. This results in an imbalance in the normal microbiome known as dysbiosis, which can be associated with many detrimental health conditions, including antibiotic-associated diarrhea (AAD), inflammation, allergic reactions, and even high levels of stress or anxiety[13,14,18–21]. Moreover, dysbiosis may also contribute to obesity and neurological disorders[20–24]. To avoid these consequences, one of the most common reasons that patients seek out probiotics is to restore the balance of their microbiome in the gut with beneficial microbes during a course of antibiotics. However, their benefits are often compromised by the antibiotic treatments themselves, which also kill the probiotic bacteria strains. Additionally, the issue of concurrent (or closely spaced) administration of antibiotics with beneficial bacteria extends beyond orally administered probiotics. The most well-studied clinical form of bacteriotherapy is fecal microbiota transplant (FMT), which is already used to treat severe *Clostridium difficile* infections and is under investigation to treat many other diseases and lifespan extension[25–29]. However, the administration of the antibiotic during the FMT therapy is extremely challenging due to the lack of precise management to avoid the overlap with the subsequent localization of probiotics.

Physical encapsulation of beneficial bacteria inside a protective shell could present an attractive approach to address the imprecise killing action of antibiotics on probiotics[29–40]. Although encapsulation of probiotics inside various polymeric particles has been explored to improve their survivability and colonization in GI tissues[41,42], a simple, safe, and rapid coating with broad-spectrum protection against antibiotics has not been addressed. Herein, we show that probiotic cells can be armored individually by a biocompatible supramolecular coating composed of tannic acids (TA) and ferric ions (Fe$^{III}$) (referred to as 'probiotic nanoarmor') without compressing their activity (Fig. 1)[43–47]. The commonly used components for this nanoarmor are mainly abundant plant extracts and already have been used as a food additive and functional materials in cell engineering[44,45,48,49], making the nanoarmor ideal for the use of probiotic engineering. Mechanistic studies revealed that molecular interaction-mediated inactivation of antibiotic molecules by the nanoarmor prevents cellular uptake and imprecise killing action. This safe and edible nanoarmor provides a series of common probiotic bacterial species with protection from antibiotics, significantly increasing their resistance against a spectrum of often-used antibiotics in vitro and in the intestinal tract of live rats, prolonging their survival ability and probiotic potential of restoring a healthy balance in the gut microbiome.

## Results

### The protection of armored probiotic bacteria from antibiotics.
*Escherichia coli* Nissle1917 (EcN), a type of commonly used probiotics for therapeutics and diagnostics, was first selected as the model bacteria to demonstrate the defense of nanoarmor against antibiotics[50]. EcN grown to mid-logarithmic phase were pelleted, then washed and resuspended in phosphate buffer saline (PBS). We constructed a series of armored probiotics by means of a biocompatible, polyphenol-based assembly method[43,45] (Supplementary Fig. S1). Confocal laser scanning microscopy (CLSM) confirmed the formation of a nanoshell surrounding EcN (Fig. 2a and Supplementary Fig. S2). Transmission electron microscopy (TEM) and scanning electron microscopy (SEM) also exhibited a uniform nanoarmor with a thickness of about 20 nm (Fig. 2b, Supplementary Figs. S3 and S4). The nanoarmor strategy could be easily applied to other probiotic bacteria, including Gram-positive strains like *Lactobacillus casei* ATCC393T (*L. casei*)[51] and commercial blends of probiotic strains (CVS Pharmacy Health Probiotic Capsules, CVS HPC, formulated with 10 different bacterial species) (Fig. 2c, d). To verify the formation of polyphenol-based nanoarmor around the bacteria, Zeta potential measurements showed that the surface charges of EcN, *L. casei*, and CVS HPC shifted to the more negative potentials after being armored (Supplementary Fig. S5), which was in line with the negative charges of polyphenol-based supramolecular network reported previously[44]. The X-ray photoelectron spectroscopy (XPS) results also supported the presence of Fe$^{III}$-TA nanoarmor around the bacteria (Supplementary Fig. S6).

We then investigated the ability of the nanoarmor to protect bacteria from the actions of antibiotics (Supplementary Fig. S7). After being subjected to the nanoarmor protection strategy, EcN samples were exposed to six different clinically relevant antibiotics at excess levels of their respective minimum bactericidal concentration (MBC, Supplementary Table S1) for 24 h. The viability of the bacteria was then evaluated using colony-forming unit (CFU) counting. CFU determination showed that the armored EcN could withstand excessive MBC levels of antibiotics, whereas the growth of naïve probiotics was inhibited completely (Fig. 2e). Armored EcN recovered from the antibiotics could grow in fresh LB culture media, whereas no growth was observed for naïve EcN under the same conditions (Supplementary Fig. S8). Moreover, the nanoarmor protection strategy was extended to *L. casei* ATCC393T and also to a commercial blend of probiotic strains (CVS HPC) containing 10 different bacterial species (Supplementary Table S2). CFU counts of armored and naïve *L. casei* ATCC393T after antibiotic exposure showed that the armored probiotics remained viable in the presence of the same six antibiotics used in the group of EcN, whereas the naïve probiotics showed no viability in the prescence of antibiotics (Fig. 2f). For the commercial blend of probiotic strains, the result of CFUs remained the same for those subjected to the nanoarmor protection (Fig. 2g). Metagenomic sequencing for the probiotic mixture showed that, in addition to preserving the number of CFUs, the nanoarmor protection also preserved the phylogenetic diversity of the bacterial consortium in the presence of antibiotics, although the relative abundance of the different strains showed minor changes[20] (Fig. 2h, i).

### Interactions between the nanoarmor and antibiotic molecules.
To get insights into the protection mechanism of nanoarmor against the antibiotic molecules, we further investigated the molecular interactions between the nanoarmor and antibiotic molecules. The six antibiotics used in the experiments have different molecular structures and mechanisms of actions, suggesting that the protective mechanism of the nanoarmor is generalizable (Supplementary Fig. S9). The Brunauer, Emmett, and Teller (BET) method of adsorption of nitrogen gas and cross-sectional TEM showed a typical microporous structure of armored probiotics, with a pore diameter ranging from 2.34 to

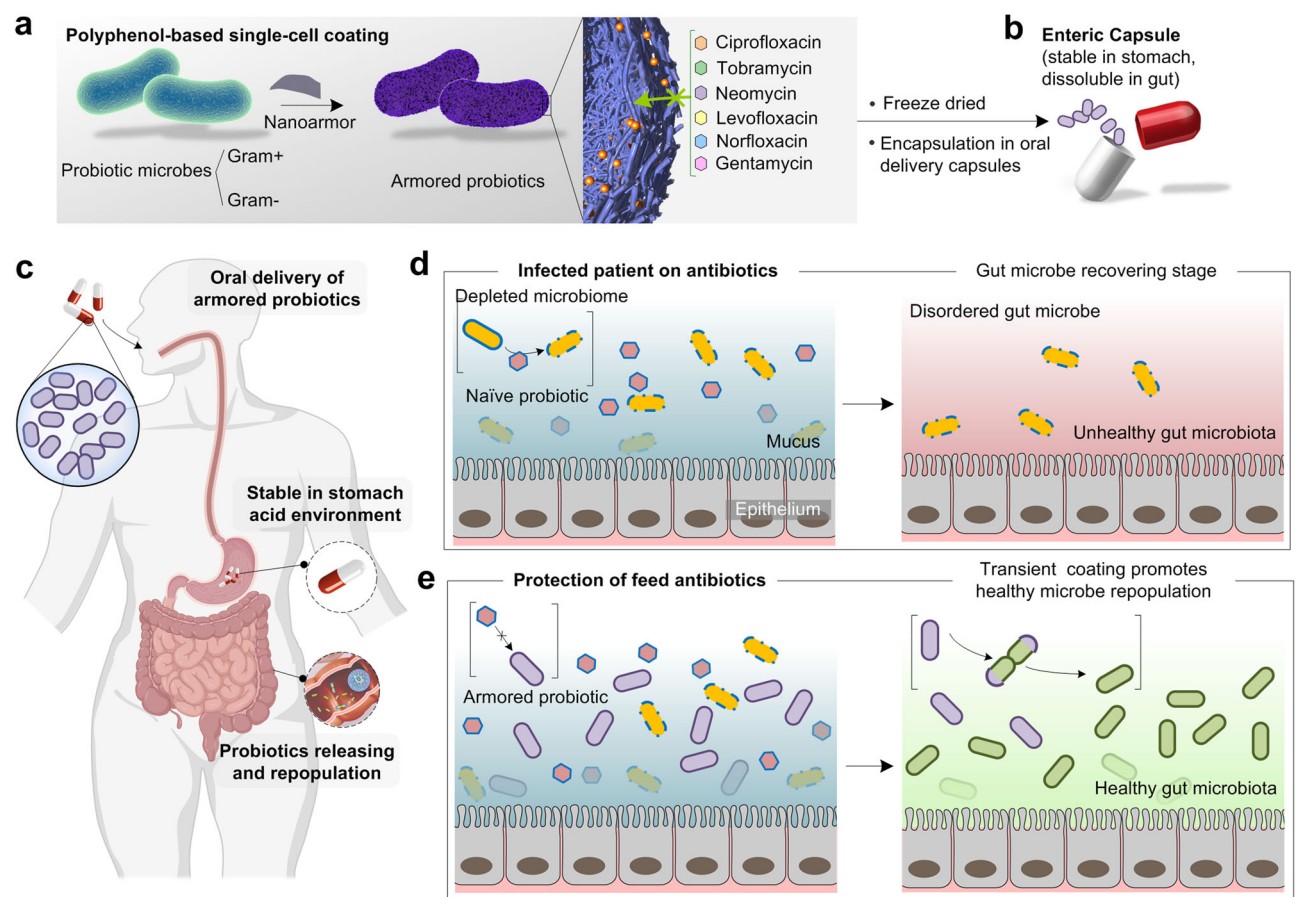

**Fig. 1 Natural polyphenol-based single-cell coating (nanoarmor) for the protection of bacteria from antibiotics in the gastrointestinal (GI) tract. a** The nanoarmor enables a rapid and highly biocompatible single-cell encapsulation that protects from a wide range of antibiotics with different molecular structures and properties. **b** Armored probiotic bacteria can be freeze-dried and filled into enteric capsules designed for oral delivery. **c** The enteric capsule remains intact during the low pH of gastric transit and releases the armored probiotics in the gut. **d** The poor specificity of antibiotics normally depletes healthy commensals in the gut and hinders probiotic treatments. **e** The nanoarmor provides a safe and transient coating to the beneficial bacteria from antibiotics, facilitating healthy microbe repopulation.

10.86 nm (Supplementary Fig. S10). These pore sizes are large enough for 200 kDa molecules to pass through[45]. This indicates that physical occlusion could not be the primary mechanism by which the armored probiotics were protected. The supramolecular structure of nanoarmor contains abundant catechol and galloyl groups, suggesting that there might be accessible interacting sites. These sites can prevent the antibiotic molecules from directly contacting the cell membrane. Our previous work demonstrated that the supramolecular network of $Fe^{III}$-TA could form versatile interactions that bind to organic surfaces and biological molecules via a range of multiple interactions[44,46]. Therefore, we hypothesize that the supramolecular nanoarmor of $Fe^{III}$-TA on the bacterial cells inactivates antibiotics via multiple intermolecular interactions that hinder their internalization into the encapsulated cells[52]. Quartz crystal microbalance (QCM) was used to monitor the interactions between the nanoarmor of $Fe^{III}$-TA surface and antibiotic molecules (Fig. 3a). When antibiotics flowed over a bare Au chip, only minor changes in absorbed mass were observed after 1,000 s of equilibration (Fig. 3b). In contrast, $Fe^{III}$-TA armored substrates showed large increases in absorbed mass after the introduction of antibiotics (Fig. 3c, d), indicating preferential interactions between antibiotics and the $Fe^{III}$-TA surface mediated by the formation of multiple molecular interactions (e.g., hydrogen bonding, hydrophobic interactions, and electrostatic interactions)[53]. To further support this mechanistic rationale, we added $Fe^{III}$-TA aggregates to each of the six

antibiotic solutions individually in the absence of bacteria for 30 min, followed by the EcN sample addition, and then cultured with Luria-Bertani (LB) for another 24 h. The EcN showed high bacterial viability, in contrast to cultures exposed to the antibiotics without $Fe^{III}$-TA pretreatments (Fig. 3e), suggesting that the antibiotics were still trapped in the $Fe^{III}$-TA aggregates, and could not inhibit the growth of EcN. In addition, we measured the antibiotic concentrations in the supernatants of the solutions treated with $Fe^{III}$-TA aggregates by high-performance liquid chromatography (HPLC). More than 90% of the antibiotics could be absorbed by the $Fe^{III}$-TA aggregates, and therefore the remaining antibiotics were not able to completely kill the probiotic bacteria (Supplementary Fig. S11). This result indicated that the protection mechanism of nanoarmor to probiotic bacteria was mainly based on the mechanism of antibiotics adsorption, creating a long-term microenvironment with low antibiotics concentration around the probiotic cells.

**Encapsulation of lyophilized bacteria and cell recovery.** To implement the nanoarmor strategy in the application of oral administration, lyophilized bacteria cells were placed in the enteric capsules (Fig. 4a) with a polymer coating (Eudragit L100) that remains intact under acidic conditions and releases the capsule contents at higher pH values encountered in the intestine[54,55]. After the lyophilization process, the naïve probiotics and armored probiotics showed similar viabilities and growth

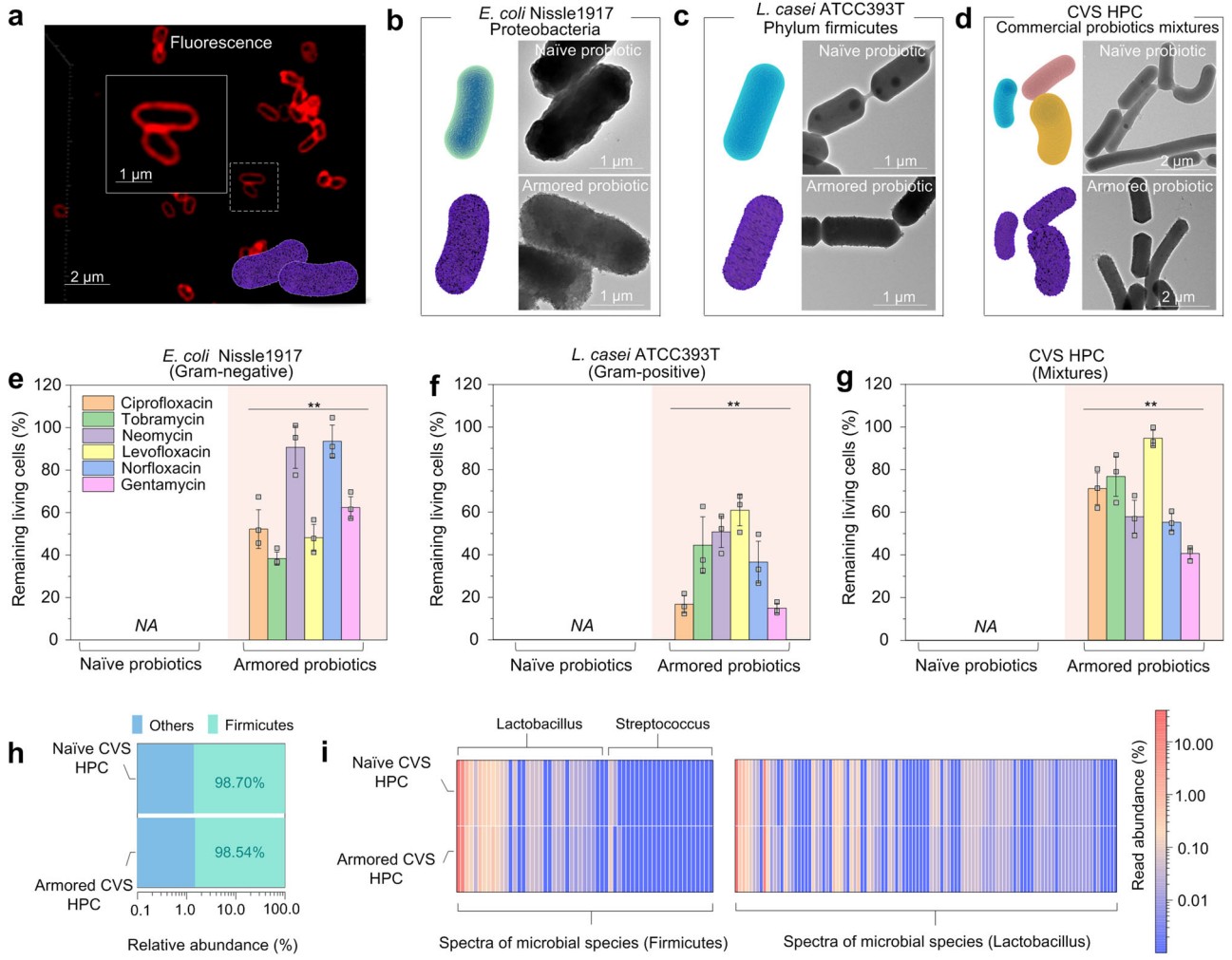

**Fig. 2 Nanoarmor of bacteria and protective effect from a wide range of antibiotics. a** CLSM images of armored EcN. Nanoarmor was labeled with bovine serum albumin conjugated with Alexa Fluor 647. **b** TEM images of naïve or armored EcN. **c** TEM images of naïve or armored *L. casei*. **d** TEM images of naïve or armored CVS HPC. **e** Relative bacterial viability of naïve or armored EcN after exposure to six different antibiotics for 24 h, *NA*, not detectable. Statistical comparisons were performed using two-tailed unpaired *t*-tests (*n* = 3). For the group of ciprofloxacin, *p* = 0.001267 < 0.01. For the group of tobramycin, *p* = 0.000064 < 0.001. For the group of neomycin, *p* = 0.000207 < 0.001. For the group of levofloxacin, *p* = 0.00041 < 0.001. For the group of norfloxacin, *p* = 0.00065 < 0.001. For the group of gentamicin, *p* = 0.000067 < 0.001. **f** Relative bacterial viability of naïve or armored *L. casei* ATCC393T after exposure to six different antibiotics for 24 h, *NA*, not detectable. Statistical comparisons were performed using two-tailed unpaired *t*-tests (*n* = 3). For the group of ciprofloxacin, *p* = 0.004041 < 0.01. For the group of tobramycin, *p* = 0.009074 < 0.01. For the group of neomycin, *p* = 0.000618 < 0.001. For the group of levofloxacin, *p* = 0.000296 < 0.001. For the group of norfloxacin, *p* = 0.00614 < 0.01. For the group of gentamicin, *p* = 0.000681 < 0.001. **g** Relative bacterial viability of commercial blends of naïve probiotic strains CVS HPC and armored blends after exposure to six different antibiotics for 24 h, *NA* not detectable. Statistical comparisons were performed using two-tailed unpaired *t*-tests (*n* = 3). For the group of ciprofloxacin, *p* = 0.000178 < 0.001. For the group of tobramycin, *p* = 0.000305 < 0.001. For the group of neomycin, *p* = 0.000435 < 0.001. For the group of levofloxacin, *p* = 0.000003 < 0.001. For the group of norfloxacin, *p* = 0.000041 < 0.01. For the group of gentamicin, *p* = 0.000024 < 0.001. **h** Taxa summary map of different phyla in naïve or armored CVS HPC. **i** Heatmap of metagenomic sequencing results of different Fimicutes strains of naïve or armored CVS HPC. **p*-value < 0.05; ***p*-value < 0.01; ****p*-value < 0.001. The graphs represent mean values ± standard error of mean for Fig. 2e-g. Significant differences between mean values were evaluated using ANOVA with multiple comparisons.

curves (Fig. 4b, c). Although natural polyphenols generally possess antibacterial capacity, the formation of nanostructured networks around the cells (Supplementary Figs. S4 and S10) show a neglected effect on the bacteria probably due to the formation of supramolecular nanocomplexes based on metal-phenolic coordination (Supplementary Fig. S12)[54]. The protection offered by the nanoarmor is inherently transient since bacteria become susceptible to the antibiotic again after cell division since it ensures that the bacteria essentially return to their original state on a relatively fixed timescale. We confirmed the feasibility of the double encapsulation scheme with a simple in vitro model that mimicked GI transit. Lyophilized EcN (armored and naïve) were

placed inside the enteric capsules and incubated first in a simulated gastric fluid (SGF) for 2 h, and then in a simulated intestinal fluid (SIF) for another 12 h, either with or without antibiotics present (Fig. 4d). The enteric capsule remained intact in the acidic SGF (pH 1.2) and released the lyophilized EcN in the SIF (pH 6.8) (Supplementary Fig. S13). In the absence of antibiotics, both armored and naïve EcN exhibited comparable viability. This result indicated that the nanoarmor could not affect the growth of the bacteria in full media. Cross-sectional TEM images indicated that the shell of nanoarmor could be shared by the divided bacteria so that the protective effect can be maintained even after the cell division (Supplementary Fig. S14). When antibiotic

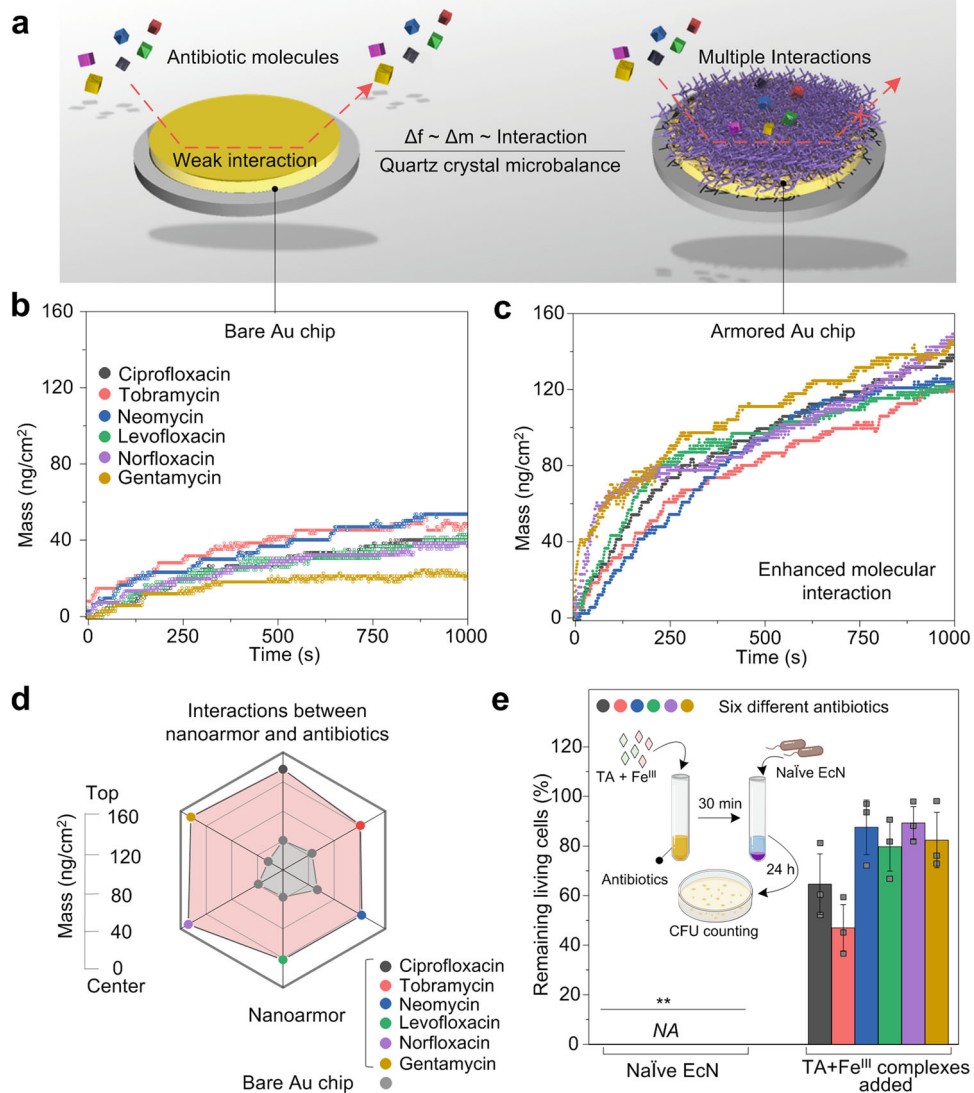

**Fig. 3 Interaction between the cell nanoarmor and antibiotic molecules. a** Schematic representation of the nanoarmor on QCM chips which enables enhanced mass adsorption due to the multiple interactions between polyphenol moieties and antibiotic molecules. The change of interfacial interactions can be detected by the frequency change, $\Delta f$, which is proportional to the mass of the absorbed molecules, $\Delta m$. **b** QCM analysis shows mass change over time as various antibiotics flowed over a bare gold substrate or **c** a substrate armored with $Fe^{III}$-TA. **d** Absorbed mass values on the bare and $Fe^{III}$-TA functionalized substrates after equilibrium. **e** Cell viability of EcN incubated with either antibiotic at their respective MBCs (*NA* not detectable) or with supernatants from each antibiotic premixed with $Fe^{III}$-TA nanocomplex. Statistical comparisons were performed using two-tailed unpaired *t*-tests ($n = 3$). For the group of ciprofloxacin, $p = 0.001713 < 0.01$. For the group of tobramycin, $p = 0.002102 < 0.01$. For the group of neomycin, $p = 0.000354 < 0.001$. For the group of levofloxacin, $p = 0.00033 < 0.001$. For the group of norfloxacin, $p = 0.000045 < 0.001$. For the group of gentamicin, $p = 0.00048 < 0.001$. *$p$-value $< 0.05$; **$p$-value $< 0.01$; ***$p$-value $< 0.001$. The graphs of Fig. 3e represent mean values ± standard error of mean. Significant differences between mean values were evaluated using ANOVA with multiple comparisons.

levofloxacin was present in the SIF, the armored EcN maintained significantly higher viability than naïve EcN (Fig. 4e). The morphological changes of armored and naïve probiotics were profiled by cross-sectional TEM. After the treatment of levofloxacin for 3, 6, or 12 h, the morphology of the armored probiotics remained intact, while deformed morphology can be observed in the naïve probiotics due to the killing action (Supplementary Fig. S15). The nanoarmors were kept intact due to the significantly slow division rate of bacteria in the simulated intestinal fluid without a culture medium. In addition, the armored EcN recovered from the SIF even in the presence of levofloxacin could also recover and reach the plateau in LB culture media, whereas no growth was observed for naïve EcN under the same condition (Fig. 4f). In vitro assays for the adhesion of naïve/armored probiotics to intestinal mucus

of rats revealed that the nanoarmor did not affect the adhesion of probiotics to intestinal mucus, which may be due to the naturally inherent mucoadhesive property of TA from the nanoarmor (Supplementary Fig. S16)[56].

**In vivo protection and biological performance of the armored probiotics**. We then examined the protection ability of the armored probiotics from the administration of antibiotics in vivo. In the first stage, healthy Wistar rats received levofloxacin (2 g/L) in their drinking water for 3 days (Fig. 5a). Subsequently, 10 mg of lyophilized *E. coli* consortium (EcC$_{tet}$, containing a modified strain BL21(DE3) with a tetracycline resistance gene and *E. coli* Nissle1917) including naïve or armored groups encapsulated by

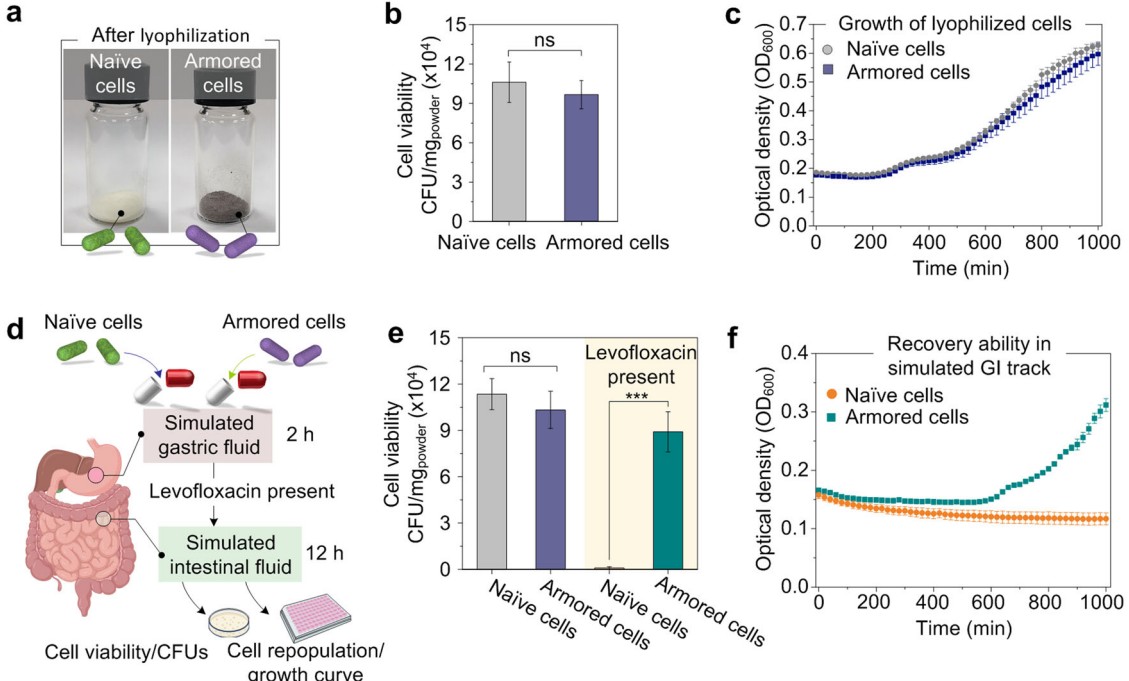

**Fig. 4 Encapsulation of lyophilized bacteria and cell recovery in simulated GI conditions. a** Images of lyophilized powders of naïve EcN and armored EcN. **b** CFU of naïve or armored EcN after lyophilization. Statistical comparisons were performed using two-tailed unpaired $t$-tests ($n = 3$), $p = 0.559161 > 0.05$. **c** Growth curve of naïve or armored EcN after lyophilization in the timescale of 1800 min. **d** Schematic of the enteric capsule filled with bacteria used for the experiments with simulated gastric and intestinal fluids. **e** CFU of naïve or armored EcN after encapsulation in enteric capsules and treatment with simulated gastric and intestinal fluids in the presence or absence of levofloxacin. Statistical comparisons were performed using two-tailed unpaired $t$-tests ($n = 3$). For the groups without levofloxacin, $p = 0.448823 > 0.05$, ($n = 3$). For the groups with levofloxacin, $p = 0.000669 < 0.001$. **f** Growth curve of naïve or armored EcN after recovery from the simulated intestinal fluid containing levofloxacin in the timescale of 1800 min. $ns$ $p$-value $> 0.05$, *$p$-value $< 0.05$; **$p$-value $< 0.01$; ***$p$-value $< 0.001$. The graphs of panels **b** and **e** represent mean values ± standard error of mean. Significant differences between mean values were evaluated using ANOVA with multiple comparisons.

enteric capsules were administrated daily through oral gavage for the next 6 days. Levofloxacin was administrated throughout the second stage. In the third stage, the administration of both bacteria and antibiotics was ceased while the experiment continued for an additional 2 days. Rats in both control groups received the naïve or armored probiotics, but without the administration of levofloxacin. Fecal samples were collected once daily in the second and third stages. The samples were homogenized and subjected to CFU counting on tetracycline-selective plates to assess the presence of $EcC_{tet}$.

For rats without the administration of levofloxacin, CFU counts of naïve and armored $EcC_{tet}$ were similar throughout the experiment, maintaining low counts. The CFU counts slowly grew from $0.2 \times 10^6$ (naïve) and $0.16 \times 10^6$ (armored) CFU/g of feces to $0.88 \times 10^6$ (naïve) and $0.52 \times 10^6$ CUF/g of feces, respectively, from day 4 to day 9, and decreased gradually after the cessation of bacterial administration (Fig. 5b). Colonization resistance of healthy gut microbiota led to the relatively low number of $EcC_{tet}$ colonies in the group without the administration of levofloxacin[20,57,58]. For the cohorts with the administration of levofloxacin, the fecal CFU counts of naïve $EcC_{tet}$ increased slowly over the course of the experiment, with a final concentration of $2.84 \times 10^6$ CFU/g of feces on the 9th day. Meanwhile, the fecal CFU counts of armored $EcC_{tet}$ flourished rapidly from $2.60 \times 10^6$ to $12.56 \times 10^6$ CFU/g. The CFU counts reached the peak number of $12.88 \times 10^6$ (armored) CUF/g of feces on day 7, and maintained stability on day 8 to day 11, indicating that the nanoarmor shells provided sufficient protection to the $EcC_{tet}$ and successfully colonized in the gut for animals receiving continuous antibiotics. Fecal concentrations of both naïve and

armored $EcC_{tet}$ showed further increase after stopping the administration of $EcC_{tet}$ and antibiotics on day 9. If the rats received levofloxacin without the administration of $EcC_{tet}$, no colonies can be observed on the tetracycline-selective plates throughout the experiments (Supplementary Fig. S17).

Moreover, the drastic depletion of beneficial gut microbiota by levofloxacin led to AAD in the experimental groups, which resulted in the weight loss of animals (Fig. 5c). Importantly, the oral administration of armored $EcC_{tet}$ reversed the trend of decline in bodyweight on the 7th day. The trend of decline in bodyweight in the naïve group can be reversed after the ceasing of antibiotic administration. The severity of AAD and dysfunction of GI tract was further examined by the assessment of fecal samples (Fig. 5d, e). Enzyme-linked immunosorbent assay (ELISA) and real-time quantitative polymerase chain reaction (RT-qPCR) assays have shown that the administration of armored probiotics could improve some of the pre-inflammatory symptoms caused by AAD (Supplementary Figs. S18–S21). In contrast to those treated with naïve probiotics, treating rats with LCB showed decreased pre-inflammatory symptoms as reflected by the lower levels of proinflammatory cytokines (interleukin-6, interleukin-1β, and tumor necrosis factor-α) and higher anti-inflammatory cytokine (interleukin-10) in serum, as well as the downregulation of genes of proinflammatory colonic cytokines in the GI tract. Treating with armored $EcC_{tet}$ also upregulated the gene of anti-inflammatory cytokine interleukin-10 and tight junction proteins (occludin, claudin-1).

The feces of animals that had not received antibiotics maintained a confined shape and dark-brown color throughout

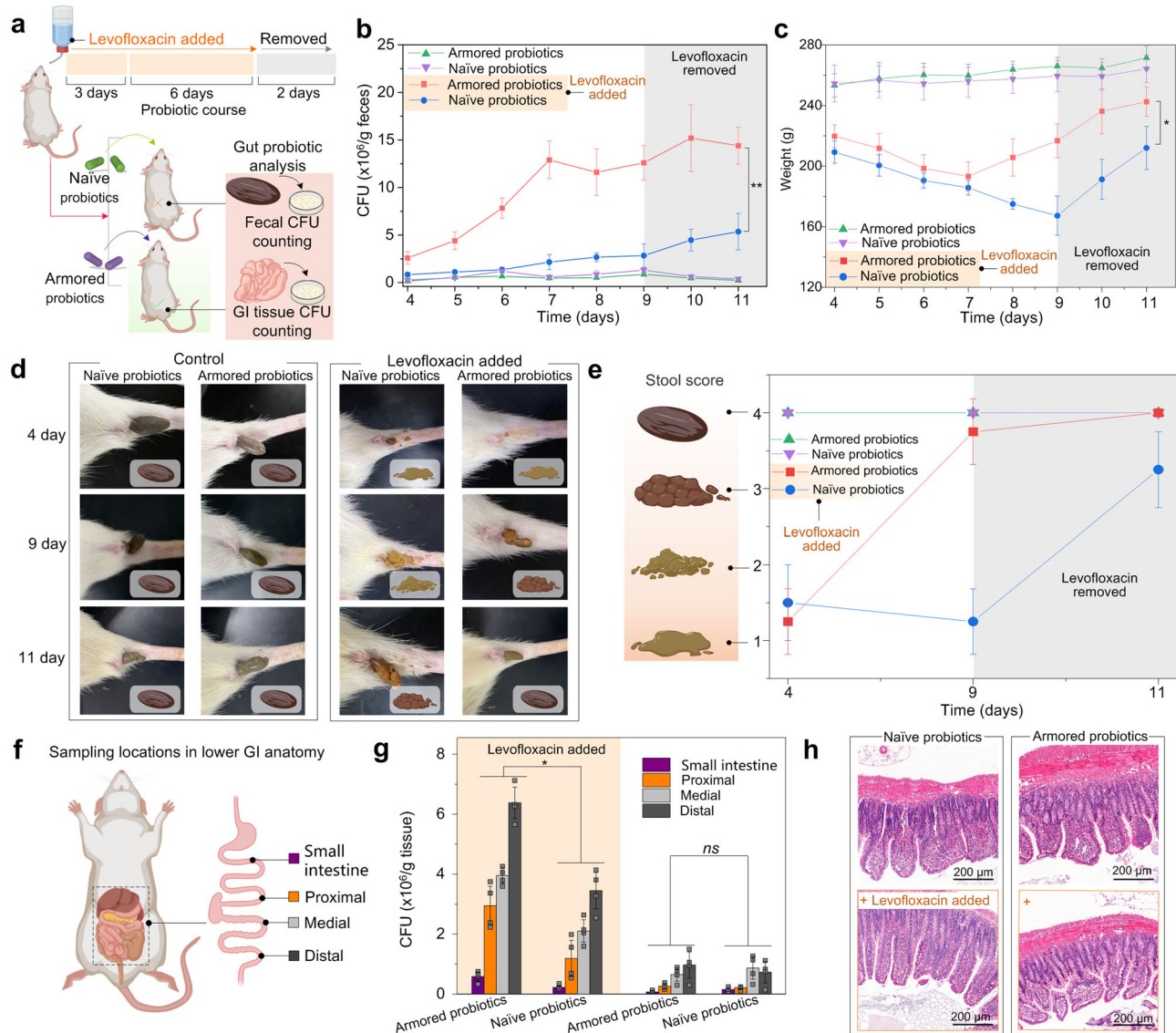

**Fig. 5 In vivo protection of armored probiotics from antibiotics. a** Schematic representation of animal experiment design. **b** Fecal CFU counts of EcC_tet concentration in rats receiving daily oral administrations of lyophilized bacteria in enteric capsules. For the two cohorts, levofloxacin (2 g/L) was present in the drinking water for the first 9 days and bacteria were administered on days 4–9, followed by two days with neither EcC_tet nor antibiotics. The other two cohorts received the same bacterial administration schedule without the antibiotics. Statistical comparisons were performed using two-tailed unpaired $t$-tests ($n = 4$), $p = 0.001211 < 0.01$. **c** Bodyweight of rats measured daily. Statistical comparisons were performed using two-tailed unpaired $t$-tests ($n = 4$), $p = 0.02189 < 0.05$. **d** The representative images of the corresponding fecal samples in the naïve or armored groups. **e** Stool score of rats measured on days 4, 9, 11. **f** Schematic of tissue sampling locations from harvested GI tracts at the endpoint. **g** CFU counts of EcC_tet from homogenates of GI tract sections. Statistical comparisons were performed using two-tailed unpaired $t$-tests ($n = 4$). For the group of small intestine with levofloxacin, $p = 0.006575 < 0.01$. For the group of proximal with levofloxacin, $p = 0.013795 < 0.05$. For the group of medial with levofloxacin, $p = 0.000398 < 0.001$. For the group of distal with levofloxacin, $p = 0.000728 < 0.001$. For the group of small intestine without levofloxacin, $p = 0.110634 > 0.05$. For the group of proximal without levofloxacin, $p = 0.412247 > 0.05$. For the group of medial without levofloxacin, $p = 0.413450 > 0.05$. For the group of distal without levofloxacin, $p = 0.487825 > 0.05$. **h** Representative histological sections obtained from the distal of each cohort, visualized with H&E stain. ns $p$-value > 0.05; *$p$-value < 0.05; **$p$-value < 0.01; ***$p$-value < 0.001. The graphs of panels **b**, **c**, and **g** represent mean values ± standard error of mean. Significant differences between mean values were evaluated using ANOVA with multiple comparisons.

the experiment compared with healthy rats, indicating that administration of naïve or armored EcC_tet showed neglected side effects (Supplementary Fig. S22). In contrast, animals that took armored EcC_tet showed fecal shape changes from watery to the normal state, and color changes from yellowish-brown to dark-brown during day 4 to day 11. Meanwhile, the shape and color of the feces treated with naïve EcC_tet also showed a slow improvement from day 4 to day 11. The feces of animals that had not received antibiotics maintained stable water contents and

Na$^+$ levels throughout the experiment (Supplementary Fig. S23). In contrast, animals that took armored EcC_tet showed a more significant decrease in fecal water contents and Na$^+$ levels from day 4 to day 11, when compared with the group treated with naïve EcC_tet. No occult blood was observed for all of the fecal samples throughout the experiment. These results were consistent with the visual fecal consistency score, providing additional evidence to support the therapeutic effects of armored EcC_tet for the AAD animals.

To determine the spatial distribution of the bacteria within the gut, the GI tracts of all groups were harvested, homogenized, and subjected to CFU counting on selective plates (Fig. 5f, g). CFU counts maintained low counts in the small intestine, but maintained high counts from proximal to distal. These results mirrored the observation of fecal pellets that the concentration of EcC$_{tet}$ was lower in the absence of antibiotic (distal, $0.96 \times 10^6$ (armored) and $0.73 \times 10^6$ (naïve) CFU/g of tissue), but was much higher in concentration in the presence of the antibiotic (distal, $6.40 \times 10^6$ (armored) and $3.48 \times 10^6$ (naïve) CFU/g of tissue). It should be noted that the endpoint data were obtained 2 days after EcC$_{tet}$ and antibiotic administration had ceased. Therefore, the difference between the observed endpoint data conditions is likely to be less than the differences on days 4–9 of the experiment. Throughout the experiment, no detrimental physiological effects were observed in any of the animals, which exhibited normal tissue morphology, as assessed by histological staining of fixed tissue sections (Fig. 5h, Supplementary Figs. S24 and S25). In the biological toxicity test, rats were orally dosed with up to 20 mg of armored EcC$_{tet}$ daily. The results showed that armored EcC$_{tet}$ did not show significant biological toxicity to the rats (Supplementary Figs. S26–S28). The results showed that the expression of proinflammatory and anti-inflammatory factors (interleukin-6, interleukin-1β, tumor necrosis factor-α, and interleukin-10) in the serum of any of the rats was normal. There was no significant difference in six biochemical indicators between the rats who received armored probiotics and the control rats. Metagenomic sequencing for the fecal bacteria of rats showed that the armored probiotics had no significant effect on the composition of the intestinal flora of healthy rats, though the relative abundance of the different strains showed minor changes (Supplementary Fig. S29).

## Discussion

In this study, we showed that a polyphenol-based single-cell coating (referred to as 'probiotic nanoarmor') composed of TA and Fe$^{III}$ can form a transient barrier on the probiotics. The dihydroxyphenyl (catechol) or trihydroxyphenyl (galloyl) of natural polyphenols were known to strongly bind to diverse surfaces through covalent and non-covalent interactions[44]. Therefore, the natural polyphenol-based nanoarmor is suitable for different probiotics including Gram-negative strains like EcN, Gram-positive strains like *L. casei*, and the commercial blend of probiotic strains.

The nanoarmor provided protection from a wide range of antibiotics with varying structures and mechanisms of action, and overcame a critical issue in the concurrent administration of antibiotics and probiotics. The nanoarmor inactivated the imprecise killing action of antibiotics by intermolecular interaction and thereby prevented cellular uptake and killing action. The polyphenol-based nanoarmor created a microenvironment with low antibiotic concentration for probiotics by absorbing various antibiotics near the armored probiotics. This protection mechanism shows a long-lasting protective effect, even after the probiotics have divided and broken through the shell of nanoarmor.

The protection offered by the nanoarmor can persist in vivo after oral administration of enteric capsules loaded with armored probiotics in order to promote the colonization of probiotics in the AAD mammalian GI tract. Despite such armored probiotics do not confer particular health benefits for normal rats, we anticipate that this strategy can be implemented to enhance the efficacy of probiotic treatment regimens where antibiotics must be administered concurrently or in close proximity to the probiotic, such as in the treatment of inflammatory bowel disease.

Moreover, it could be useful in the context of other medical procedures involving therapeutic bacteria, such as FMT, curated commensal consortia, or engineered bacteria, where antibiotic administration must be managed carefully.

## Methods

All experiments were conducted in accordance with US National Institutes of Health guidelines and approved by the Experimental Animal Center of Sichuan University.

**General materials, bacterial strains, and animals**. TA was purchased from Sigma-Aldrich (USA). Ferric(III) chloride hexahydrate (FeCl$_3$·6H$_2$O) was purchased from Chron Chemical Co., LTD (China). LB culture and MRS culture were purchased from Hopebio (China). Ciprofloxacin, tobramycin, neomycin, levofloxacin, norfloxacin, and gentamicin were purchased from Aladdin (China). PBS buffer was purchased from Adamas life (China). All of these materials were used as received. TNF-α, IL-6, IL-10, and IF-1β ELISA kit were obtained from Mlbio, China. High-purity Milli-Q (MQ) water with a resistivity of 18.2 MΩ cm was obtained from an inline Millipore RiOs/Origin water purification system.

*E. coli* Nissle1917 was purchased from Biobw (China). The tetracycline resistance *E. coli* BL21(DE3) was a kind gift from the Fei Wang Lab (Chengdu Institute of Biology). *L. casei* ATCC393T was purchased from China Center of Industrial Culture Collection (CICC, China). CVS Health Probiotic Capsules (CVS HPC) were purchased from CVS. For all growth steps, unless otherwise noted, *E. coli* Nissle1917 and BL21(DE3) were grown in lysogeny broth (LB) medium at 37 °C. Starter cultures were grown to a density of $1 \times 10^8$ CFU/mL before being diluted to $1 \times 10^5$ CFU/mL (5 mL) for experiments. Similar protocols were applied to analyze the growth of *L. casei* and the CVS HPC mixture, using MRS broth medium and growing at 37 °C in the sealed vials to crudely approximate anaerobic conditions. Male 8-week-old Wistar rats were purchased from Dashuo Laboratory Animal Technology, Ltd. (China).

Rats were housed in SPF conditions with sterile food and water *ad libitum*. Rats were maintained in sterile vinyl isolators equipped with food, water, and bedding in the Sichuan University West China Medical Center animal facility. Before any experiment, mice had at least 1 week to acclimatize to the facility environment.

**Instruments**. 3D-reconstructed fluorescence microscopy imaging was performed using a Leica SP5 X MP inverted confocal microscope equipped with a 60 × 1.42 NA oil immersion objective, with a set of standard filters for DAPI/CFP/FITC/AF488/AF568/Cy5/AF647. Image processing and 3D models were analyzed and generated with Imaris (Bitplane) software using the maximum intensity projection. UV-Vis absorption and fluorescence measurements were conducted on an Infinite M Nano microplate reader (Tecan Group, Switzerland). Transmission electron microscopy (TEM) was performed on a Tecnai G2 F20 S-TWIN TEM instrument, operating at a voltage of 100 kV (FEI USA, Inc.). Scanning electron microscopy (SEM) was performed on a JEOL JSM-7500F SEM instrument. Particle zeta potential was measured by dynamic light scattering (DLS) on Zetasizer Nano ZSP (Malvern, UK). Ultrathin sections (about 80 nm) were cut on a Reichert Ultracut-S microtome, picked up onto copper grids stained with lead citrate, and examined in a JEOL 1200EX Transmission electron microscope, and images were recorded with an AMT 2k CCD camera.

**The preparation of armored probiotics**. The preparation of armored probiotics was based on our reported literatures[40,41,45]. Probiotics (EcN, *L. casei*, or CVS HPC) ($2 \times 10^5$ CFU/mL) were washed with PBS three times and then suspended in 600 µL PBS solution. 50 µL FeCl$_3$·6H$_2$O (1.25 mg/mL) and 50 µL tannic acid (5 mg/mL) solutions were added into the cell suspension. Finally, 300 µL PBS was added. Ten seconds of vortexing were required between each addition.

**Fluorescent images**. The armored probiotics were labeled by exposuring upon the solution of bovine serum albumin conjugated to Alexa-647 (BSA-Alexa-647, 5 mg/mL) for 20 min. The stained, armored probiotics were mounted onto a glass slide and coverslip using Prolong Diamond Antifade mountant. Bacterial cells were imaged using a Zeiss TIRF/ LSM 710 confocal microscope.

**TEM microscopy sample preparation**. Cells were firstly fixed with glutaraldehyde. The samples were then dehydrated in an alcohol gradient (50–70–95–100%). 10.0 µL of biohybrid suspensions were allowed to air-dry on the formvar carbon-coated copper grids to prepare samples for TEM observation.

**BET method of adsorption of nitrogen gas**. Nitrogen adsorption-desorption measurements were carried out on a 3Flex surface characterization analyzer (Micromeritics Instrument Corporation, U.S.A.). Before measurement, all samples were degassed at 100 °C under vacuum for 12 h. The pore parameters such as specific surface area, pore size distribution, and pore volume were obtained according to the nitrogen adsorption-desorption isotherms at 77 K. The specific surface area values of these samples were acquired by using the BET method. The

pore size distribution profiles from the nitrogen adsorption branch of isotherms were obtained by applying a non-local density functional theory (NLDFT) and a carbon slit pore model. The non-negative regularization is 0.01. Pore volumes were calculated from the amount of nitrogen adsorbed at the relative pressure of 0.97.

**ELISA assay.** The serum of rats was collected to measure the levels of interleukin-6, interleukin-1β, tumor necrosis factor-α, and interleukin-10 (IL-6, IL-10, TNF-α, and IL-1β) using their respective kits according to the manufacturer's instruction (Mlbio, China).

**RT-qPCR assay.** Intestinal tissues of rats were collected. Total RNAs were extracted by Triquick Reagent (Trizol Substitute, Solarbio, China). RNA (500 ng), quantified by NanoDrop2000 (Thermo Fisher Scientific), was reversely transcribed to cDNA using the first-strand cDNA synthesis kit (Vazyme, China). Quantitative PCR was applied using the SYBR Green dye (Vazyme, China) on quant studio 3 applied biosystems (Thermo Fisher Scientific). All primers were synthesized by Tsingke Biotechnology and their sequences were listed in Supplementary Table S3. The parameters of PCR assays were shown as following: initial denaturation at 95 °C for 30 s, 40 cycles of denaturation at 95 °C for 10 s, and primer annealing and reaction at 60 °C for 30 s. Comparative quantification was assessed using $2^{-\Delta\Delta Ct}$ method with glyceraldehyde-3-phosphate dehydrogenase (GAPDH) as the endogenous control.

**The adherence of armored probiotics to intestinal mucus of rats.** The intestine of SD rats weighing about 200 g was taken, and the contents were gently washed with PBS (pH 7.4). The mucus was gently scraped with a slide on an ice bath. Then the mucus was combined and homogenized in pre-chilled PBS, and was centrifuged (5000 r/min, 30 min). The supernatant was extracted by adding two times the volume of anhydrous ethanol and placed in a refrigerator at −20 °C overnight to allow the mucus to precipitate and lyophilize. The precipitate was dissolved in carbonate buffer (pH 9.6) at 0.1 g/(100 mL). One hundred fifty microliters of rat intestinal mucus was added to the well of a 96-well plate. The 96-well plate was incubated overnight at 4 °C, and then the plate was blocked with 200 μL of PBS (with 1% Tween 20) at room temperature for 1 h. Two hundred microliters of the armored/naïve EcN suspension was added to the mucus-coated wells. The wells were incubated for 4 h at 4 °C. Then the wells were washed twice with PBS, and were fixed with ethanol for 10 min. The adhered probiotics were stained with pink or Hoechst, and were measured at OD540 or observed under a fluorescence microscope.

**Ultrathin section TEM sample preparation.** Naïve and armored probiotics were centrifuged at $1000 \times g$ for 2 min and the pellet was resuspended in 5 μL of 20% BSA. The cell/BSA mixture was dispensed on the 100 μm side of a type A 6 mm Cu/Au carrier (Leica), covered with the flat side of a type B 6 mm Cu/Au carrier (Leica), and frozen in a high-pressure freezer (EM ICE, Leica). The samples were freeze substituted at −80 °C for 48 h in an automated freeze substitution device (AFS2; Leica) in acetone containing 1% $H_2O$, 1% $OsO_4$, and 0.1% uranyl acetate. The temperature was increased 5 °C per hour up to 20 °C and the samples were rinsed several times in acetone at room temperature. The samples were infiltrated with Spurr's resin (EMS) mixed with acetone 1:1 overnight at 4 °C and moved to embed molds filled with freshly mixed Spurr's resin at room temperature.

**SEM sample preparation.** Cells were fixed with glutaraldehyde. The samples were then dehydrated in an alcohol gradient (50–70–95–100%). 10.0 μL of biohybrid suspensions in MQ water were allowed to air-dry on silicon wafers which were cleaned with acetone, ethanol, and MQ water. All the samples were treated with spray-gold before observation.

**MBC assay.** Bacteria (EcN, *L. casei*, CVS HPC) were cultured in respective cultures (LB or MRS) for 24 h at 37 °C under shaking of 150 rpm. After centrifugation at $3000 \times g$ for 5 min, bacteria in the culture medium were collected and resuspended in PBS to $2 \times 10^5$ CFU/mL as the working suspension. Antibiotics (ciprofloxacin, tobramycin, neomycin, levofloxacin, norfloxacin, and gentamicin) were diluted to concentrations ranging from 3.125 μg/mL to 800 μg/mL by a two-fold gradient dilution in a 96-well plate. After mixing equal volumes of bacterial cell suspension (50 μL) and antibiotic solution (50 μL) in each well, the 96-well plates were incubated at 37 °C for 24 h. The MBC values were identified as the lowest drug concentration to kill over 99.9% of bacteria. An aliquot of 10 μL bacterial suspension from each well was plated onto LB agar plates. After the plates were incubated at 37 °C for 24 h, MBC values were determined by visual inspection of CFU on the agar.

**Metabolic activity of armored probiotics after antibiotic treatments.** The viability of the armored probiotics was measured by CFU counts. EcN, *L. casei*, and CVS HPC cells were washed with ultrapure water three times and then diluted to $2 \times 10^5$ CFU/mL with PBS. Then 100 μL of bacterial suspensions were added to each well of a 96-well plate (Nest, China). Ten microliters $FeCl_3 \cdot 6H_2O$ (1.25 mg/mL) solution, 10 μL tannic acid (TA) (5 mg/mL) solution, and 60 μL of PBS were added

into the cell suspension. Finally, 10 μL antibiotics (ciprofloxacin, tobramycin, neomycin, levofloxacin, norfloxacin, gentamicin) at their respective MBC final concentrations and 10 μL bacterial culture (LB, MRS) were added. Ten seconds of vortexing were required between each addition. The mixture was cultured at 37 °C for 24 h. After that, harvested bacteria were then plated on LB or MRS agar plates and their viability was evaluated by CFU counting using a serial dilution method. CFU counting was performed to quantitatively assess bacteria viability.

**Bacterial growth curves.** Armored bacteria were first treated with antibiotics for 24 h as described above. Then, the cells were washed three times via centrifugation and resuspension in PBS to remove the antibiotic. The growth of the washed cells was measured by resuspending in LB media in a 96-well plate and monitoring OD600 using a microplate reader. Measurements were taken every 20 min for 15 h at 37 °C.

**Metagenomic sequencing.** DNA degradation degree and potential contamination were monitored on 1% agarose gels. DNA concentration was measured using Qubit® dsDNA Assay Kit in Qubit® 2.0 Fluorometer (Life Technologies, CA, USA). OD value was controlled between 1.8 and 2.0, DNA contents above 1 μg are used to construct the library. A total amount of 1 μg DNA per sample was used as input material for the DNA sample preparations. Sequencing libraries were generated using NEBNext® Ultra™ DNA Library Prep Kit for Illumina (NEB, USA) following the manufacturer's recommendations and index codes were added to attribute sequences to each sample. Briefly, the DNA sample was fragmented to a size of 350 bp by sonication, then DNA fragments were end-polished, A-tailed, and ligated with the full-length adaptor for Illumina sequencing with further PCR amplification. At last, PCR products were purified (AMPure XP system) and libraries were analyzed for size distribution by Agilent2100 Bioanalyzer and quantified using real-time PCR. The clustering of the index-coded samples was performed on a cBot Cluster Generation System according to the manufacturer's instructions. After cluster generation, the library preparations were sequenced on an Illumina Novaseq 6000 platform and paired-end reads were generated. Genes with a *p*-value less than 0.05 were considered to be significantly differentially expressed. A system for assigning taxonomic labels to short DNA sequences was processed using Kraken2.

**Zeta potential of armored probiotics.** The zeta potential of the naïve and armored EcN, *L. casei*, and CVS HPC was measured after suspending the cells in 500 μL ultrapure water at an OD_600 of 0.4. Measurements were taken using a Zetasizer (Malvern, UK) instrument.

**XPS characterization of armored probiotics.** The incident radiation of X-ray photoelectron spectroscopy (XPS, Thermo Fisher Corporation, USA) was monochromatic Al Kα X-rays (1486.6 eV) at 220 W (22 mA and 10 kV). Survey (wide) and high-resolution (narrow) scans were recorded at analyzer pass energies of 100 and 50 eV, respectively. Survey scans were obtained using a step size of 1.0 eV and a dwell time of 100 ms. Narrow high-resolution scans were run over a 20 eV binding energy range with a 0.05 eV step size and a 250 ms dwell time. Dried cell samples (armored/naïve) were used for the XPS measurements.

**QCM with dissipation.** The amount of antibiotics captured by the nanoarmor was evaluated by depositing the $Fe^{III}$-TA on the surface of the Au chip, then flowing a solution of the appropriate antibiotic over the coated surface and evaluating absorbed mass using a qCell T Q2 (3 T analytik, 3 T GmbH & Co. KG, Germany) quartz crystal microbalance. The dynamic tests were carried out at a constant flow rate of 60 μL/min. The mass of the immobilized antibiotics was calculated by the Kelvin−Voigt model.

**Pre-incubation of polyphenol and antibiotics before bacteria exposure.** In order to assess the adsorptive effects of polyphenols in preventing antibiotic toxicity to bacteria, 70 μL of antibiotics (at their respective MBC final concentrations) were first incubated with 10 μL $FeCl_3$ (2.5 mg/mL) solution and 10 μL TA (10 mg/mL) solution in the absence of bacteria. After 30 min, the concentration of antibiotics in the supernatant was tested by HPLC and the adsorption rate of antibiotics by nanoarmor was calculated. Then, 100 μL the EcN ($2 \times 10^5$ CFU/mL) sample was added to the mixture and cultured with 10 μL LB culture at 37 °C for 24 h. Then the bacteria were plated on LB agar plates and their viability was evaluated by CFU counting using a serial dilution method to quantitatively assess bacterial viability.

**In vitro test with simulated gastric fluid and simulated colonic fluid.** Armored EcN and naïve EcN cells were lyophilized to powders, and then filled into Eudragit L100 coated capsules (size type 9 h, TORPAC inc, USA). The capsules were transferred into a simulated gastric fluid (16.4 mL HCl, 800 mL $H_2O$, 10 g pepsin) for 2 h and later moved into a simulated intestinal fluid (6.8 g $KH_2PO_4$, 500 mL $H_2O$, 0.1 mol/L NaOH adjust pH to 6.8, 10 g trypsin; for the experiments on the presence of antibiotics, added levofloxacin at 6.25 μg/mL of final concentration) for 12 h. The higher pH of the simulated intestinal fluid led to the dissolution of the

Eudragit capsule and the release of the lyophilized bacterial cells inside. The cells obtained from the ruptured capsules were directly used for the measurement of cell viability through CFU counting.

**Cytotoxic tests**. NIH3T3 cells were maintained in DMEM (Invitrogen, Carlsbad, CA, USA) containing 10% fetal calf serum (Invitrogen) and 1% penicillin/streptomycin at 37 °C in a 5% $CO_2$ atmosphere. A total of $2 \times 10^5$ NIH3T3 cells were plated in a six-well plate for 24 h and then treated with 100 μg/mL of nanoarmor as the final concentration for 24 h at 37 °C. After incubation, the cells were harvested and washed with ice-cold PBS. The apoptosis ratio was performed with an annexin V-FITC Apoptosis Detection Kit (Beyotime) using a Gallios flow cytometer (Beckman Coulter). ten microliters of the cell suspension of each group was observed by an inverted biological microscope (Olympus, CKX53).

**Animal experiments**. Male 8-week-old Wistar rats were purchased from Dashuo Laboratory Animal Technology, Ltd. (China). The rats were randomly grouped into four rats per group. Rats were given drinking water with levofloxacin (2 g/L) for 3 days. Subsequently, the rats were given daily oral administrations of Armored E. coli consortium (EcC$_{tet}$, contained a modified strain BL21(DE3) with a tetracycline resistance gene and EcN). For administration, 10 mg of the EcC$_{tet}$ that had been previously armored were lyophilized and filled into Eudragit L100 coated enteric capsules (size type 9 h, TORPAC inc, USA). The enteric capsules containing bacteria were administered daily to each rat by oral gavage for 6 days, during which time the drinking water contained levofloxacin. On the 10th day, the rats were given drinking water free of antibiotics and the oral capsule administrations ceased. Bodyweight was recorded daily for each rat, including two days after bacterial administration had ceased. Fecal samples were collected once a day on days 4–11 and plated on tetracycline-selective agar. CFU (colony-forming units) counts were obtained. Obtained the fecal photograph on days 4, 9, 11, and record the relevant stool score. After 11 days, the rats were sacrificed and their gastrointestinal tracts were harvested, gently washed with PBS (pH 7.4) to clean the contents, homogenized, and subjected to CFU counting on selective plates to determine the spatial distribution of the bacteria within the gut.

As for the fecal parameters, the fecal samples of rats were collected daily and were detected using the Modified Fecal Occult Blood Test Kit (EZ Detect). Next, the fecal samples were freeze-dried. The water content of the feces was calculated from the weight change before and after drying. The freeze-dried fecal samples were analyzed for $Na^+$ levels by inductively coupled plasma mass spectrometry (ICP-MS, 7700, Agilent Technologies) after digestion.

In terms of biosafety of armored probiotics, healthy Wistar rats received 0/10/20 mg of lyophilized armored EcC$_{tet}$ encapsulated by enteric capsules daily through oral gavage for the 6 days. The Blood biochemical parameters were measured in all groups. H&E staining was used to analyze the toxicity of armored EcC$_{tet}$ on the major organs of rats.

**Statistical analyses**. The relevant experiments presented in the current study were performed independently at least three times. Statistical tests were calculated in Microsoft Excel. The details of the statistical tests carried out are indicated in respective figure legends. The graphs represent mean values ± standard error of mean. Significant differences between mean values were evaluated using ANOVA with multiple comparisons. $p$-values were computed for two- or three-way ANOVA. Rats were randomized in different groups before being assayed. TEM, SEM, and fluorescent observations were repeated three times independently with similar results. H&E staining experiments were repeated three times independently with similar results. No data were excluded in the final statistical analysis.

**Reporting summary**. Further information on research design is available in the Nature Research Reporting Summary linked to this article.

## Data availability

The data supporting the findings from this study are available within the article file and its supplementary information. Metagenomic sequencing data generated in this study have been deposited in the NCBI database under accession code PRJNA741919 and in the GSA database under accession code CRA005726. Any other raw data or non-commercial material used in this study are available from the corresponding author. Source data are provided with this paper.

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

## Acknowledgements
The work in the J.G. laboratory was financially supported by the National Global Talents Recruitment Program (J.G.), National Natural Science Foundation of China (J.G., Grant No. 22178233), State Key Laboratory of Polymer Materials Engineering (J.G, Grant No. sklpme 2020-3-01), Double First Class University Plan (J.G.), Key Laboratory of Leather Chemistry and Engineering (J.G.), National Engineering Research Center of Clean Technology in Leather Industry (J.G.), The Fundamental Research Funds for the Central Universities (J.S., Grant No. YJ201959), Science and Technology Support Program of Sichuan Province (J.S., Grant No. 2021YJ0290; Y.H. Grant No. 2021YJ0414) and China Postdoctoral Science Foundation (Y.H., Grant No. 2020TQ0209). We would like to thank S. Wang at the Analytical & Testing Centre of Sichuan University for TEM characterization; Elizabeth Benecchi and Maria Ericsson at Harvard Medical School electron microscopy facility for help on microtome techniques. We would like to thank Kelly Ibsen (Harvard University) for her help in the design of the animal experiments. We would like to thank Biorender for the help in creating images.

## Author contributions
J.G., J.P., G.G., C.C. and N.S.J. conceived of the idea. D.B. conducted the experiments of the confocal microscope. Y.Z. performed the metagenomic sequencing and the corresponding data analysis. G.G., Y.L., and Z.J. planned and designed the animal experiments. Y.Z., J.S., Q.W., and Y.H. assisted with the animal experiments. J.P., G.G., J.G., N.S.J., and C.C. drafted the manuscript. All authors discussed the results and commented on the manuscript.

## Competing interests
The authors declare no competing interests.
