## [Peer Review File · Nature Communications]

REVIEWER COMMENTS

Reviewer #1 (Remarks to the Author):

In this manuscript, Pan et al. developed a strategy for probiotics oral administration using a single-cell coating composed of natural polyphenols, which could protect bacteria from the action of antibiotics. The authors demonstrated that the multiple interactions between the coating and antibiotic molecules allow the antibiotics to be effectively absorbed onto the nanoarmor. Moreover, armored probiotic bacteria were freeze-dried and filled into enteric capsules for oral delivery and the enteric capsule remained intact during the low pH of gastric transit and released the armored bacteria in the gut. However, similar strategies to protect probiotics from antibiotics have been reported extensively. In addition, why did not take antibiotics-absorbent substances orally to reduce the concentration of antibiotics and then deliver probiotics to regulate gut disorders caused by antibiotics as the antibiotics-absorbent coating may cause high local concentration of antibiotics, which may further damage the bioactivity of probiotics. Compared with other coatings with physical barrier properties, the data in this manuscript does not show the unique benefits and advantages of the nanoarmor. Besides, the data obtained is insufficient to support the conclusions, especially in the verification of the beneficial effects of coated probiotics *in vivo*. Furthermore, there is a bit confusion in Figure 3 and 4 (see detailed comments). Therefore, the novelty and significance of the manuscript is insufficient to merit publication in *Nature Communications*. It could be suitable for a more specialized journal after addressing following issues.

1. The authors pointed out the view that physical occlusion could not be the primary mechanism by which the armored bacteria were protected as previous work on the polyphenol-based coating of FeIII-TA network has indicated that the average pore sizes of the mesh-like network are large enough for 200 kDa molecules to pass through. However, there are many protein and polysaccharide structures on the surface of bacteria, which are different from smooth chemical particles. Therefore, the authors need to measure the average pore sizes of the mesh-like network to prove this point.

2. The authors should apply more characterization methods to prove that the coating can adsorb antibiotics. In addition, the authors did not directly test the effect of coating on probiotic activity and shape after adsorbing antibiotics. There is also no data showing the enteric capsules can remain intact during the low pH of gastric transit and release the armored bacteria in the gut. The author emphasized that the nanoarmor is a temporary coating that will fall off during the proliferation of bacteria, but there is no direct evidence to support this. In Figure 4f, the armored EcN did not recover from SIF in the presence of levofloxacin in the first 10 hours, which may indicate the antibiotics absorbed onto the nanoarmor could damage probiotics.

3. The authors should detect more intestinal pathological indicators to evaluate the therapeutic effect of the coated probiotics, rather than just use fecal shape and color as the single and relatively subjective indicator to judge the treatment efficacy. In addition, the level of inflammatory factors in the serum should be included in the biosafety test.

4. The coating with physical barrier properties can hinder the penetration of a variety of stimulating factors, including gastric juice, bile salts and antibiotics, thereby protecting the activity of probiotics in the intestine, which can excellently solve the problem of antibiotics affecting the efficacy of probiotics, while the armored coating can only specifically adsorb antibiotics and cannot resist irritating ingredients in the intestine, such as bile salts. Please elaborate on the advantages of this coating in oral probiotics.

5. Given polyphenol is a bacterial inhibition or killing material, is there any effect on bacteria viability after decorating with polyphenol?

6. Details to prepare this antibiotic-resistant probiotics should be included in the experimental section.

7. In figure 1a, as E.coli belong to Gram-, Lact belong to Gram+, it should not be presented probiotic microbes like the way indicated in this paper.

8. In Figure 2a, there is no control for bacteria with dye only to ensure the fluorescent signal is truly from nanoarmor.

9. Fig5-g, why the bacterial counts were lower when there was no antibiotics applied?

10. The numbers of how many mice were used in each group and the replicates of each in vivo/vitro experiments were missed in all the figures.

Reviewer #2 (Remarks to the Author):

The authors used polyphenol-based armored probiotics to provide protection against antibiotics that are used simultaneously or after antibiotic therapy. In vitro experiments showed protection from antibiotics while naïve unprotected probiotics did not. In rat challenge experiments, researchers also showed the ability of armored probiotics to withstand antibiotic challenge, survive, flourish and prevent antibiotic-induced diarrhea. Though the approach seems interesting, the lack of mechanistic insights of protection prevents it from fully evaluate its broad utility.

Major Experimental Suggestions

1. Polyphenols including tannic acid used in this study are inhibitory to many bacteria. Some probiotic bacteria such as those used in this study may be resistant; however, many gut bacteria are susceptible. What was the effect of armored bacteria on the gut microbiota of rats?
2. As stated by the authors, armored protection is only good on the parental strain. However, it is surprising to note that the dividing cells also survive and grow in the presence of antibiotics in in vitro experiments (Figure 4e and 4f)! Any explanation?
3. During antibiotic therapy, antibiotics are maintained at a certain dose level in the body. If the daughter cells do not have armored protection, how well do these probiotics survive in the gut in the presence of antibiotics for colonization and persistence to exert health beneficial effects?
4. Due to inconsistent and unpredictable beneficial effects of probiotics, bioengineering strategies have been introduced to assist probiotics to interact with specific host cell receptors to protect against infectious pathogens (for example, see a recent report by Drolia et al. 2020. Nat. Commun. 11, 6344). In such an application, researchers expressed heterologous bacterial proteins on the surface of probiotics. To broaden the utility of nanoarmor technology, authors need to demonstrate that the nanoarmor does not interfere with bacterial surface protein expression and their adhesion and colonization capabilities in vitro and in vivo.
5. Authors reported that EcCtet survived and replicated in the rat intestine in the presence of levofloxacin. Increased survival and growth of EcCtet in the rat gut could be due to their cross-resistance to levofloxacin. It is generally known resistance to one antibiotic can increase bacterial resistance to another. Have the cross-resistance patterns of this strain to levofloxacin been evaluated? Increased resistance of EcCtet may explain the increased survival capacity of the dividing probiotic strain over time to levofloxacin!

6. Immunological response and health parameters of rats after feeding with armored probiotic has not been fully evaluated to assess any negative effect on health. Inflammatory and immunomodulatory responses need a full evaluation.

7. Lack of rigor in diarrhea scoring: Only visual fecal consistency score was used however, other parameters as to water loss, Na⁺ levels, hemocult test, etc should strengthen the claim.

8. (Figure 2). The authors indicate that armored *E. coli* Nissle (EcN) recovered from the antibiotics could also be grown in fresh LB culture media, whereas no growth was observed for naïve EcN under the same conditions (Figure S7). However, antibiotic treatments are known to induce sublethal stress in bacteria. Did the authors observe any antibiotic-induced sublethal stress or injury on the armored probiotics? Such stress may reduce the survival of probiotics and eventually their beneficial attributes.

Minor comments

1. Page 14: “We then examined the protection ability of the armored antibiotic from the administration of antibiotics in vivo.” How did you coat the antibiotic with polyphenol?

2. Fig 5g. Do the bacterial counts represent total counts from luminal content and tissues or tissues alone? If it is from tissues, were the tissues washed before analysis to determine the levels of colonized probiotics counts? Were those probiotics verified to be the parental strain that was introduced, not the resident probiotic strains that have acquired resistance?

3. Page 6 + 9: “The XPS results also supported the presence of polyphenol-based nanoarmor around the bacteria (Figure S5).” What is FeIII TA? Define at its first appearance!

4. Method: The detail of the nanoarmor preparation is scanty to be reproduced by others!

5. Page 18: “The metabolic activity of the armored bacterial cells was measured by CFU counts” – How do you measure metabolic activity by CFU? A misleading statement. Viable but nonculturable cells (VBNC) are metabolically active but do not grow on agar plates without proper activation.

6. Line numbering of the text would be very helpful during the review process

Reviewer #3 (Remarks to the Author):

This paper is a very well written paper and outlines a novel approach to live encapsulated bacteria to address the negative impacts of antibiotic induced depletion of gut microbiota. Overall there is extensive experimental data to support the major findings.

I have some comments the authors may consider to improve clarity overall.

- The title of the paper doesn't adequately capture the main observation of the manuscript. I would suggest a change to the title to emphasize the major findings of the study and its potential role as a therapeutic strategy. The term safe and transient are also vague, lacking scientific objectivity and are not demonstrated in a mechanistic sense in this study

- The paper repeatedly refers to the 'inherently transient' nature of the coating. I believe the authors should provide outline a more mechanistic basis of the time frame involved in dissolution of the coating in the intestine – this is important to understand the impact clinically. Apparently the coating protects the probiotics but also apparently doesn't reduce its ability to replicate so I feel a discussion around the duration/kinetics of the coating would be helpful

- A key observation of the study is that the therapeutic benefits of the encapsulated probiotics are only evident in a diseased condition. I am not a fan of the term 'probiotics' as it is widely misinterpreted that probiotics are health promoting. This study confirms that the encapsulated bacteria provide a therapeutic benefit to address an iatrogenic disease condition but in fact also demonstrates that administration in a healthy condition does not influence confer improvements in gut microbial composition. The fact that the treatment is only beneficial in the disease condition needs to be very clear throughout. For example the claim that 'armoured probiotics have shown the ability to maintain a steady state concentration inside the gastrointestinal tract' is not factually correct. What is steady state concentration? Is there a defined steady state level of microbiota? This study demonstrated it it speeds up the recovery due to antibiotic treatment but it did not maintain equivalent levels to healthy controls. The authors need to openly acknowledge that such probiotics do not confer any particular health benefits - and are only for a specific disease condition

- Other comments include avoiding non scientific terms such as

 - o notoriously imprecise would be better to be nonspecific in their nature

 - o claims that dysbiosis contribute to diabetes need to be supported with scientific studies. To my knowledge while there is a link between altered my microbiome and disease conditions the suggestion that they contribute to this needs to be supported with scientific evidence

o also the argument that probiotics replenish their microbiome this is maybe not an accurate interpretation of the scientific literature. Probiotics are there to improve diversity or restore imbalances

- Methods: the study builds on a previous study referenced in 37 and 38 but I believe the methods and experimental detail on the coating procedures to encapsulate in the nano armor should be detailed in this manuscript. While the schematic is useful it lacks methodological detail. This should be provided to provide the reader with the opportunity to replicate this procedure
- This study demonstrates that in figure 5G treatment produces levels of cfu in the intense time which are higher than the healthy scenario. The authors claim that their approach is 'safe' but have they discussed the potential risks where potentially excessive colonization of the intestine occurs in these treatment groups

Bold text indicates reviewer critique

Blue text indicates responses or manuscript revisions. All new inclusions in the revised manuscript are highlighted in *red*.

Our responses to the specific requests made by the three reviewers are as follows.

Response to Reviewer #1:

In this manuscript, Pan et al. developed a strategy for probiotics oral administration using a single-cell coating composed of natural polyphenols, which could protect bacteria from the action of antibiotics. The authors demonstrated that the multiple interactions between the coating and antibiotic molecules allow the antibiotics to be effectively absorbed onto the nanoarmor. Moreover, armored probiotic bacteria were freeze-dried and filled into enteric capsules for oral delivery and the enteric capsule remained intact during the low pH of gastric transit and released the armored bacteria in the gut.

Q1-1: However, similar strategies to protect probiotics from antibiotics have been reported extensively.

RI-1: We thank the reviewer for raising this point on the difference between the polyphenol-based nanoarmor and previously reported physical barriers. We agree that considerable developments have been made in the bacterial microencapsulation field owing to the potential and broad scope of applications of encapsulated probiotics. However, only a few articles have reported the protection of bacteria against one or two antibiotics by physical isolation mainly *in vitro* (please see references below, *RI-1_Ref 1-Ref 3*). Therefore, the demonstration of cytoprotective and therapeutic effects of protected probiotics *in vivo* is still highly worthwhile to study and contributes a significant progress to the field, as evaluated by the Reviewers #2 and #3.

In addition, when the bacteria divide and breakthrough this protective shell, the bacteria inside the capsule are re-exposed to the cytotoxic antibiotic environment. The conventional protection strategy based on physical barriers show limitations to be applied in patients who require long-term administration of antibiotics. In this work, the polyphenol-based nanoarmor creates a microenvironment with low antibiotic concentration for probiotics by absorbing various antibiotics near the armored probiotics. This protection mechanism has not been reported previously and shows a long-lasting protective effect, even after the probiotics have divided and broke through the shell of nanoarmor.

Finally, from the perspective of therapeutic outcomes, our results show that the armored probiotics can colonize the intestine of AAD rats in the chronic presence of antibiotics and improve the physiological condition of AAD rats. None of the previously reported protection strategies have presented these therapeutic outcomes *in vivo*.

RI-1_Ref 1: Wang, X. *et al.* Bioinspired oral delivery of gut microbiota by self-coating with biofilms. *Sci. Adv.* **6**, eabb1952 (2020).

R1-1_Ref 2: Li, Z. A. *et al.* Biofilm-inspired encapsulation of probiotics for the treatment of complex infections. *Adv. Mater.* **30**, 1803925 (2018).

R1-1_Ref 3: Cao, Z. *et al.* Biointerfacial self-assembly generates lipid membrane coated bacteria for enhanced oral delivery and treatment. *Nat. Commun.* **10**, 1-11 (2019).

Q1-2: In addition, why did not take antibiotics-absorbent substances orally to reduce the concentration of antibiotics and then deliver probiotics to regulate gut disorders caused by antibiotics as the antibiotics-absorbent coating may cause a high local concentration of antibiotics, which may further damage the bioactivity of probiotics.

R1-2: We thank the reviewer for bring up this discussion. We agree that taking antibiotics-absorbent substances orally to reduce the concentration of the antibiotics in the gut might be a promising strategy to regulate gut disorders caused by antibiotics administration. However, the antibiotic-absorbent substances also reduced the uptake of antibiotics by intestinal, which would compromise the therapeutic purpose of oral antibiotics. In our work, the polyphenol-based nanoarmor will only absorb antibiotics around the coated probiotics, creating a cytoprotective microenvironment with low antibiotic concentration for the probiotics. Therefore, we obtain the rationale that the strategy of nanoarmor shows advantageous aspects compared to the proposed assumption of antibiotic absorbents.

Q1-3: Compared with other coatings with physical barrier properties, the data in this manuscript does not show the unique benefits and advantages of the nanoarmor.

R1-3: Thanks to the reviewer for his or her pointing out the comparison. Actually, our polyphenol-based nanocoating strategy was inspired from the pioneering works on bacterial physical barriers. These previous works provided many fundamental guidance for the design and methodology of our work. While it is still necessary and valuable to develop alternative strategy for the probiotics protection. We have believe that some unique benefits of our polyphenol-based nanocoating can generate enough differentiation compared with the previously reported physical barriers.

Specifically, the absorption mechanism of polyphenol-based nanoarmor creates a long-term cytoprotective microenvironment to the probiotics, even though the bacteria divide. Moreover, our work presents the first therapeutic demonstration of protected probiotics to the antibiotic-associated diarrhea (AAD) animal experiments, which yet been reported previously. Finally, the use of FDA-approved GRAS compound of natural polyphenol and the easy preparation process enable the present work with great potential for large-scale translation and real-world applications.

To increase the readability of this manuscript, we have performed additional experiments and further strengthened our discussions to highlight these advantageous properties of our polyphenol-based armored probiotics. Please see our detailed responses in the specific questions.

Q1-4: Besides, the data obtained is insufficient to support the conclusions, especially in the verification of the beneficial effects of coated probiotics *in vivo*.

RI-4: We thank the reviewer for the valuable suggestion on therapeutic effects. We have conducted a series of additional experiments to further support our conclusion on the beneficial effects of armored probiotics *in vivo*. Please see our response in details on **RI-12** and **RI-13**, specifically including new supplementary **Figures S18, S19, S20, S21, S26, S27, S28** and their corresponding discussions in the revised manuscript.

Q1-5: Furthermore, there is a bit confusion in Figure 3 and 4 (see detailed comments). Therefore, the novelty and significance of the manuscript is insufficient to merit publication in Nature Communications. It could be suitable for a more specialized journal after addressing following issues.

RI-5: Thanks for the reviewer's comments on highlighting our novelty. We have conducted additional experiments and provided new insightful discussion to strengthen our novelty of this work. Please see our detailed point-by-point responses below. We believe that the novelty and significance of our work have been significantly improved based on the reviewer's comment.

Q1-6: The authors pointed out the view that physical occlusion could not be the primary mechanism by which the armored bacteria were protected as previous work on the polyphenol-based coating of Fe^{III}-TA network has indicated that the average pore sizes of the mesh-like network are large enough for 200 kDa molecules to pass through. However, there are many protein and polysaccharide structures on the surface of bacteria, which are different from smooth chemical particles. Therefore, the authors need to measure the average pore sizes of the mesh-like network to prove this point.

RI-6: We have conducted additional experiments according to the suggestion from the reviewer. The pore sizes of naïve/armored probiotics were measured and presented as new **Figure S10**. The new experimental results showed a typical microporous structure of armored probiotics, with a pore diameter range from 2.34 – 10.86 nm (new **Figure S10a**). This result was consistent with the observation of cross-sectional TEM (new **Figure S10b**). Particularly, the cross-sectional TEM showed that the nanoarmor was constructed by surface-assembled nanoparticles with interparticle spacing of ~ 10 nm.

To increase the readability of our manuscript, the following sentences have been added in Line 181 on Page 9 of the revised manuscript to describe the additional experiments for the measurement of pore sizes of nanoarmor, and the corresponding methods have been described in the section of materials and methods:

“The Brunauer, Emmett and Teller (BET) method of adsorption of nitrogen gas and cross-sectional TEM showed a typical microporous structure of armored probiotics, with a pore diameter ranging from 2.34 – 10.86 nm (Figure S10). These pore sizes are large enough for 200 kDa molecules to pass through⁴³.”

The following figure was added as **Figure S10** in the revised supplementary materials with the related experimental results.

Figure S10. (a) BET result of armored probiotics, which showed a typical microporous structure with a pore diameter ranging from 2.34 – 10.86 nm. (b) The cross-sectional TEM showed that the nanoarmor was constructed by the assembled nanoparticles which densely distributed on the cell surface with gaps of up to 10 nm.

Q1-7: The authors should apply more characterization methods to prove that the coating can absorb antibiotics.

RI-7: We thank the reviewer for the suggestion. We have conducted additional characterizations on the adsorption ability of the nanoarmor for antibiotics. We tested the removal of the six antibiotics by nanoarmor through high performance liquid chromatography (HPLC) (new **Figure S11**). The results showed that the nanoarmor was able to absorb > 90% of the antibiotics with the concentration of MBC in solution. This makes the residual antibiotics in the solution insufficient to kill the probiotics.

The following text has been added in Line 207 on Page 10 in the revised manuscript to enhance the discussion on the adsorption ability of nanoarmor:

*“In addition, we measured the antibiotic concentrations in the supernatants of the solutions treated with Fe^{III}-TA aggregates by high performance liquid chromatography (HPLC). More than 90% of the antibiotics could be absorbed by the Fe^{III}-TA aggregates, and therefore the remaining antibiotics were not able to completely kill the probiotic bacteria (**Figure S11**). This result indicated that the protection mechanism of nanoarmor to probiotic bacteria was mainly based on the mechanism of antibiotics adsorption, creating a long-term microenvironment with low antibiotics concentration around the probiotic cells.”*

The following figure was added as new **Figure S11** in the revised supplementary materials:

Figure S11. (a) Scheme of assays for the antibiotic adsorption ability of nanoarmor. (b) The adsorption effect of nanoarmor on six antibiotics. (c) The nanoarmor owns the ability to absorb antibiotics by the multiple interactions between the TA and antibiotics. Nanoarmor can absorb more than 90% of antibiotics and create a long-term microenvironment with low antibiotics concentration around the probiotic cells. Variation is represented by the standard deviation of three independent replicates in all graphs.

Q1-8: In addition, the authors did not directly test the effect of coating on probiotic activity and shape after absorbing antibiotics.

RI-8: According to the reviewer’s suggestion, we carried out the cross-sectional TEM to observe the morphology of naïve/armored probiotics after absorbing antibiotics in the **Figure 4e** of the original manuscript and new **Figure S15** in the revised supplementary materials. After 3 hours of treatment, the morphology of naïve probiotics showed an obvious deformation due to the antibiotic effect. In contrast, the armored probiotics were treated in simulated intestinal fluid with levofloxacin for 12 hours and still maintained the native and intact morphology. This result demonstrated the effective protection of probiotics by nanoarmor in an antibiotic environment.

The following text has been added in Line 253 on Page 13 in the revised manuscript to clarify the effect of coating on probiotic activity and shape after absorbing antibiotics:

“The morphological changes of armored and naïve probiotics were profiled by cross-sectional TEM. After the treatment of levofloxacin for 3, 6, or 12 hours, the morphology of the armored probiotics remained intact, while deformed morphology can be observed in the naïve probiotics due to the

killing action (**Figure S15**). The nanoarmors kept intact due to the significantly slow division rate of bacteria in the simulated intestinal fluid without culture medium.”

The following figure has been added as **Figure S15** in the revised supplementary materials:

Figure S15. The cross-sectional TEM images of naïve/armored probiotics after treatment of simulated intestinal fluid with levofloxacin for different times. The nanoarmors kept intact due to the significantly slow division rate of bacteria in the simulated intestinal fluid without culture medium. Scale bar, 200 nm.

Q1-9: There is also no data showing the enteric capsules can remain intact during the low pH of gastric transit and release the armored bacteria in the gut.

R1-9: Enteric capsules have been commercialized for nearly decade and widely used for the delivery of biologically active cargos to the intestinal tract (please see references below, *R1-9_Ref 1-Ref 3*). The commercial instruction of enteric capsules states that they do not dissolve in gastric juice and only disintegrate and dissolve in intestinal fluid.

Still we photographed the morphological changes of the enteric capsules after the treatment in simulated gastric/intestinal fluid (new **Figure S13**). The results showed that the enteric capsules did not dissolve in simulated gastric fluid and disintegrated rapidly in simulated intestinal fluid.

R1-9_Ref 1: Astra Zeneca, Losec® capsules-Summary of product characteristics, <https://www.medicines.org.uk/emc/medicine/7275#EXCIPIENTS>, 2017.

R1-9_Ref 2: <https://www.capsugel.com/biopharmaceutical-products/vcaps-enteric-capsules>.

R1-9_Ref 3: S. Sharma & V.R. Sinha. Liquid nanosize emulsion-filled enteric-coated capsules for colon delivery of immunosuppressant peptide AAPS. *Pharm Sci. Tech.* **19**, 881-885 (2018).

According to the reviewer's suggestion, the following text has been amended in Line 246 on Page 13 in the revised manuscript to clarify the property of the commercial enteric capsules:

“The enteric capsules remained intact in the acidic SGF (pH 1.2) and released the lyophilized EcN in the SIF (pH 6.8) (Figure S13).”

The following figure has been added as **Figure S13** in the revised supplementary materials:

Figure S13. *The photographs of the enteric capsules after treatment in simulated gastric/intestinal fluid. The enteric capsules did not dissolve in simulated gastric fluid and disintegrated rapidly in simulated intestinal fluid.*

Q1-10: The author emphasized that the nanoarmor is a temporary coating that will fall off during the proliferation of bacteria, but there is no direct evidence to support this.

R1-10: Yes, the nanoarmors do not affect the growth of the bacteria in media. **Figure 4b, c** showed that both armored and naïve EcN exhibited comparable viability in the absence of antibiotics. In addition, thanks to the helpful suggestion from the reviewer, the new cross-sectional TEM images showed that the armored EcN could divide and the shell of nanoarmor was shared by the divided bacteria so that the cytoprotective microenvironment can be maintained even after the cell division. (new **Figure S14**). Actually, this new result and discussion provide an additional evidence to support the difference between the present polyphenol-based nanoarmor and previously reported cellular physical barriers which only deliver short-term protection effect when the cell dose not divide.

The following text has been added in Line 248 on Page 13 in the revised manuscript to clarify the remaining protective effect of nanoarmor after the division of probiotic cells:

“This result indicated that the nanoarmor could not affect the growth of the bacteria in full media. Cross-sectional TEM images indicated that the shell of nanoarmor could be shared by the divided bacteria so that the protective effect can be maintained even after the cell division (Figure S14).”

The following figure has been added as **Figure S14** in the revised supplementary materials:

Figure S14. *The cross-sectional TEM image of EcN divides and breaks through the shell of nanoarmor. The nanoarmor could not affect the growth of the bacteria in media, which may be because the nanoarmor was simply shed off after the division of bacteria.*

Q1-11: In Figure 4f, the armored EcN did not recover from SIF in the presence of levofloxacin in the first 10 hours, which may indicate the antibiotics absorbed onto the nanoarmor could damage probiotics.

RI-11: We thank the reviewer for bring up this discussion. According to the suggestion from the reviewer, we repeated the experiments in the original manuscript **Figure 4e,f** for a longer time (changed from 1000 min to 1800 min). The new experimental results in the new **Figure 4e,f** showed that although the growth of armored probiotics was slightly inhibited due to the minor effect of remaining unabsorbed antibiotics, they were still able to reach plateau at 1800 min. In contrast, the naïve probiotics were completely inhibited in the antibiotic environment with no growth. This result indicates that the nanoarmor can provide effective protection to the probiotic bacteria in the antibiotic environment.

The following text has been changed in Line 259 on Page 13 in the revised manuscript to clarify the protection of nanoarmor against antibiotics:

“In addition, the armored EcN recovered from the SIF even in the presence of levofloxacin could also recover and reach plateau in LB culture media, whereas no growth was observed for naïve EcN under the same condition.”

The **Figure 4c** and **4f** have been replaced with new additional experimental results in the revised manuscript:

Figure 4. Encapsulation of lyophilized bacteria and cell recovery in simulated GI conditions. (a) Images of lyophilized powders of naïve *EcN* and armored *EcN*. (b) CFU of naïve or armored *EcN* after lyophilization. (c) Growth curve of naïve or armored *EcN* after lyophilization in the timescale of 1,800 min. (d) Schematic of the enteric capsule filled with bacteria used for the experiments with simulated gastric and intestinal fluids. (e) CFU of naïve or armored *EcN* after encapsulation in enteric capsules and treatment with simulated gastric and intestinal fluids in the presence or absence of levofloxacin. (f) Growth curve of naïve or armored *EcN* after recovery from the simulated intestinal fluid containing levofloxacin in the timescale of 1,800 min. The variation is represented by the standard deviation of three independent replicates in all graphs, *** vs the paired groups without nanoarmor, p -value < 0.001. ns, non-significant.

Q1-12: The authors should detect more intestinal pathological indicators to evaluate the therapeutic effect of the coated probiotics, rather than just use fecal shape and color as the single and relatively subjective indicator to judge the treatment efficacy. In addition, the level of inflammatory factors in the serum should be included in the biosafety test.

RI-12: We thank the reviewer for his/her concern on the use of morphological and color changes of fecal samples as indicators to prove the therapeutic effect. According to the reviewer's suggestions, we have conducted additional biomolecular characterizations to support the therapeutic effect of armored probiotics on AAD rats, including the measurements of gene expressions and cytokines determined by quantitative real-time PCR (RT-qPCR) assays and enzyme linked immunosorbent assay (ELISA) (new **Figures S18-S21**).

Specifically, *EcC_{tet}*-treated rats showed a reduced levels of proinflammatory cytokines in serum, including interleukin-6 (IL-6), interleukin-1 β (IL-1 β), and tumor necrosis factor- α (TNF- α) (new **Figures S18**), when compared with control group treated with naïve *EcC_{tet}*. Moreover, *EcC_{tet}*-treated rats also promoted the expression of anti-inflammatory cytokine interleukin-10 (IL-10) in the serum (new **Figures S18**). In addition, after the treatment of armored *EcC_{tet}*, rats showed a decrease in the expression of pro-inflammatory cytokines (IL-6, IL-1 β , TNF- α) and an increase in the expression of

anti-inflammatory cytokines (IL-10) in the serum at days 9, 11 (new **Figures S19**). The RT-qPCR assays have shown similar results. At gene expression level, the administration of armored EcC_{tet} downregulated the genes of pro-inflammatory colonic cytokines in the GI tract, including IL-6, IL-1 β , and TNF- α (new **Figures S20**). The treatment by the administration of armored EcC_{tet} also upregulated the gene of anti-inflammatory cytokine (IL-10) and tight junction proteins (Occludin, Claudin-1). After treatment with armored EcC_{tet}, rats downregulated the genes of pro-inflammatory colonic cytokines (IL-6, IL-1 β , TNF- α) and upregulated anti-inflammatory cytokine (IL-10) and tight junction proteins (Occludin, Claudin-1) at days 9, 11 (new **Figures S21**). Collectively, these results indicated that the administration of armored probiotics could improve some of the pre-inflammatory symptoms that AAD caused.

These additional experimental results and important discussions have been added in Line 316 on Page 16 in the revised manuscript to further support the therapeutic effect of the armored probiotics:

*“Enzyme linked immunosorbent assay (ELISA) and real-time quantitative polymerase chain reaction (RT-qPCR) assays have shown that the administration of armored probiotics could improve some of the pre-inflammatory symptoms caused by AAD (**Figures S18-S21**). In contrast to those treated with naïve probiotics, treating rats with LCB showed decreased pre-inflammatory symptoms as reflected by the lower levels of proinflammatory cytokines (interleukin-6, interleukin-1 β , and tumor necrosis factor- α) and higher anti-inflammatory cytokine (interleukin-10) in serum, as well as the downregulation of genes of pro-inflammatory colonic cytokines in the GI tract. Treating with armored EcC_{tet} also upregulated the gene of anti-inflammatory cytokine interleukin-10 and tight junction proteins (Occludin, Claudin-1).”*

The following figures were added as **Figures S18-S21** in the revised supplementary materials with the related experimental results:

Figure S18. Interleukin-6 (IL-6), tumor necrosis factor- α (TNF- α), interleukin-1 β (IL-1 β), interleukin-10 (IL-10) levels in serum of rats on day 11 measured by ELISA. In contrast to AAD rats (rats with the administration of levofloxacin) treated with naïve *EcC_{1et}*, treating rats with armored *EcC_{1et}* reduced the levels of proinflammatory cytokines in serum, including IL-6, IL-1 β , and TNF- α . Treating with armored *EcC_{1et}* also promoted the expression of anti-inflammatory cytokine IL-10 in the serum. Rats without the treatment of levofloxacin did not show inflammatory responses. Variation is represented by the standard deviation of three independent replicates in all graphs.

Figure S19. IL-6, TNF- α , IL-1 β , IL-10 levels in serum of AAD rats with the administration of armored *EcC_{iet}* on days 4, 9, 11 measured by ELISA. After treatment with armored *EcC_{iet}*, rats showed a decrease in the expression of proinflammatory cytokines (IL-6, IL-1 β , TNF- α) and an increase in the expression of anti-inflammatory cytokines (IL-10) in the serum at days 9, 11. Variation is represented by the standard deviation of three independent replicates in all graphs.

Figure S20. RT-qPCR analyses of IL-1 β , IL-6, TNF- α , IL-10, Occludin, and Claudin-1 mRNA levels on day 11. In contrast to AAD rats treated with naive *EcC_{iet}*, administration of armored *EcC_{iet}*

downregulated the genes of pro-inflammatory colonic cytokines in the GI tract, including IL-6, IL-1 β , and TNF- α (Figure S). Treating with armored *EcC_{let}* also upregulated the gene of anti-inflammatory cytokine (IL-10) and tight junction proteins (Occludin, Claudin-1). Rats without the treatment of levofloxacin did not show inflammatory responses. Variation is represented by the standard deviation of three independent replicates in all graphs.

Figure S21. RT-qPCR analyses of IL-1 β , IL-6, TNF- α , IL-10, Occludin, and Claudin-1 mRNA levels of AAD rats with the administration of armored *EcC_{let}* on days 4, 9, 11. After treatment with armored *EcC_{let}*, rats downregulated the genes of pro-inflammatory colonic cytokines (IL-6, IL-1 β , and TNF- α), upregulated anti-inflammatory cytokine (IL-10), and tight junction proteins (Occludin, Claudin-1) at days 9, 11. Variation is represented by the standard deviation of three independent replicates in all graphs.

Q1-13: In addition, the level of inflammatory factors in the serum should be included in the biosafety test.

R1-13: We thank the reviewer for the concern of biosafety of armored probiotics. We have added additional characterizations to profile the safety of armored probiotics in details, including blood biochemical parameters and H&E staining. The additional results showed that the expression of pro-inflammatory and anti-inflammatory factors (IL-6, IL-1 β , TNF- α , IL-10) in the serum of all rats were at normal range, and the blood biochemical indexes in the serum of all the rats were in the normal range (new **Figure S26, S27**). There was no significant difference in all indicators between the rats who received armored probiotics and the control rats. Throughout the experiment, no detrimental physiological effects were observed in any of the animals, which exhibited normal tissue morphology, as assessed by histological staining of fixed tissue sections (new **Figure S28**). the results suggested that armored probiotics presented reliable biosafety.

This important discussion has been added in Line 369 on Page 19 in the revised manuscript to clarify the biosafety of the armored probiotics:

“In the biological toxicity test, rats were orally dosed with up to 20 mg of armored EcC_{let} daily. The results showed that armored EcC_{let} did not show significant biological toxicity to the rats (Figures S26-S28). The results showed that the expression of pro-inflammatory and anti-inflammatory factors (interleukin-6, interleukin- 1β , tumor necrosis factor- α , and interleukin-10) in the serum of any of the rats was normal. There was no significant difference in six biochemical indicators between the rats who received armored probiotics and the control rats.”

The following figures were added as **Figure S26-S28** in the revised supplementary materials with the related experimental results:

Figure S26. IL-6, TNF- α , IL- 1β , IL-10 levels in serum of rats with the administration of armored EcC_{let} daily for 6 days. The results showed that the expression of pro-inflammatory and anti-inflammatory factors (IL-6, IL- 1β , TNF- α , IL-10) in the serum of any of the rats was normal. There was no significant difference in all indicators between the rats who received armored probiotics and the control rats. Variation is represented by the standard deviation of three independent replicates in all graphs.

Figure S27. Biochemical analysis of rats with different administration of armored *EcC_{1e1}*: (a) aspartate aminotransferase (AST), (b) alanine aminotransferase (ALT), (c) albumin (ALB), (d) glucose (Glu), (e) urea, (f) creatinine (CREA). There was no significant difference in all indicators between the rats who received armored probiotics and the control rats. Variation is represented by the standard deviation of three independent replicates in all graphs.

Figure S28. Representative histological sections obtained from the tissues of each cohort, visualized with H&E stain. No detrimental physiological effects were observed in any of the animals. Scale bars, 200 μm .

Q1-14: The coating with physical barrier properties can hinder the penetration of a variety of stimulating factors, including gastric juice, bile salts and antibiotics, thereby protecting the activity of probiotics in the intestine, which can excellently solve the problem of antibiotics affecting the efficacy of probiotics, while the armored coating can only specifically absorb antibiotics and cannot resist irritating ingredients in the intestine, such as bile salts. Please elaborate on the advantages of this coating in oral probiotics.

RI-14: We have divided the questions into three parts to answer separately according to the contexts of the reviewer's comments.

1) The main function of gastric juice for microorganisms is to inactivate swallowed microorganisms in the stomach, thereby inhibiting infectious agents from reaching the intestine (please see references below, *RI-14_Ref 1*). Stomach acid is neutralized in the duodenum, and thus the gastric juice-caused damage to oral probiotics can only occur in the stomach. However, in our work, the armored probiotics were delivered directly to the intestine through the use of enteric capsules so no contact with gastric juice (please see our detailed response about enteric capsules in *RI-9*).

2) Bile is a normal secretion with a regulatory effect on the intestinal flora. The human intestinal flora (including *Lactobacillus*, *Bifidobacterium*, *Enterococcus*, *Clostridium*, etc.) can be regulated by the presence of bile through the action of bacterial bile salt hydrolase (BSH) enzymes. The intestinal tract actually maintains the healthy homeostasis of intestinal flora through such regulation and avoids the over-colonization of intestinal bacteria (please see references below, *RI-14_Ref 2-Ref 5*). Therefore, it can be rationalized that the presence of bile dose not affect the therapeutic function of armored probiotics.

3) We agree that considerable developments have been made for bacterial encapsulation. However, only a few literatures have reported the protection of bacteria against one or two antibiotics by physical isolation *in vitro* (please see references below, *RI-14_Ref 6-Ref 8*). In addition, when the bacteria divide and breakthrough this protective shell, the bacteria inside the capsule are re-exposed to the antibiotic molecules. This makes this physical barrier-based approaches difficult to adopt in patients who require long-term administration of antibiotics. The polyphenol-based nanoarmor can absorb various types of antibiotics around probiotics, creating a microenvironment with low antibiotic concentration for probiotics. This adsorption of nanoarmor remains effective even after the probiotics have divided (**Figure 3e**, and new **Figure S11, S15**). Two additional advantages can be ascribed to the aspect of translation and manufacturing, including the use of natural polyphenols as FDA-approved food additives and the simple, rapid preparation process.

According to the helpful comments from the reviewers, we have added the following sentences in Line 211 on page 10 and in Line 248 on page 13 of the revised manuscript to highlight the advantageous properties of the polyphenol-based nanoarmor:

“This result indicated that the protection mechanism of nanoarmor to probiotic bacteria was mainly based on the mechanism of antibiotics adsorption, creating a long-term microenvironment with low antibiotics concentration around the probiotic cells.”

*“This result indicated that the nanoarmor could not affect the growth of the bacteria in full media. Cross-sectional TEM images indicated that the shell of nanoarmor could be shared by the divided bacteria so that the protective effect can be maintained even after the cell division (**Figure S14**).”*

Moreover, the following two literatures 37 and 38 have been added in the revised manuscript to highlight the different mechanism of bacterial protection reported by the previous works.

*37 Li, Z. A. et al. Biofilm-inspired encapsulation of probiotics for the treatment of complex infections. *Adv. Mater.* **30**, 1803925 (2018).*

*38 Cao, Z. et al. Biointerfacial self-assembly generates lipid membrane coated bacteria for enhanced oral delivery and treatment. *Nat. Commun.* **10**, 1-11 (2019).*

*RI-14_Ref 1: Sarker S A, Gyr K. Non-immunological defence mechanisms of the gut. *Gut* **33**, 987-993 (1992).*

R1-14_Ref 2: Wahlström, A. *et al.* Intestinal crosstalk between bile acids and microbiota and its impact on host metabolism. *Cell Metab.* **24**, 41-50 (2016).

R1-14_Ref 3: Derrien, M., & van Hylckama Vlieg, J. E. Fate, activity, and impact of ingested bacteria within the human gut microbiota. *Trends in Microbiol.* **23**, 354-366 (2015).

R1-14_Ref 4: Long, S. L., Gahan, C. G., & Joyce, S. A. Interactions between gut bacteria and bile in health and disease. *Mol. Aspects. Med.* **56**, 54-65 (2017).

R1-14_Ref 5: Ridlon, J. M. *et al.* Bile acids and the gut microbiome. *Curr. Opin. Gastroen.* **30**, 332 (2014).

R1-14_Ref 6: Wang, X. *et al.* Bioinspired oral delivery of gut microbiota by self-coating with biofilms. *Sci. Adv.* **6**, eabb1952 (2020).

R1-14_Ref 7: Li, Z. A. *et al.* Biofilm-inspired encapsulation of probiotics for the treatment of complex infections. *Adv. Mater.* **30**, 1803925 (2018).

R1-14_Ref 8: Cao, Z. *et al.* Biointerfacial self-assembly generates lipid membrane coated bacteria for enhanced oral delivery and treatment. *Nat. Commun.* **10**, 1-11 (2019).

Q1-15: Given polyphenol is a bacterial inhibition or killing material, is there any effect on bacteria viability after decorating with polyphenol?

R1-15: We thank the reviewer for the comment. The results in **Figure 4b,c** showed that the naïve and armored probiotics had similar viabilities and growth curves. These results indicated that polyphenol-based nanoarmor has neglected effect on the growth of armored probiotics. Moreover, a recent report also shows that the polyphenol-based supramolecular nanocoating have no cytotoxicity to the bacteria (*J. Am. Chem. Soc.* 2021, 10.1021/jacs.1c09018). This is probably due to the formation of nanostructured networks around the bacteria through metal-coordination rather than soluble molecules of polyphenols.

The following text has been added in Line 235 on Page 12 in the revised manuscript to clarify the neglected effect of nanoarmor on probiotics:

“Though natural polyphenols generally possess antibacterial capacity, the formation of nanostructured networks around the cells (Figures S4, S10) show neglected effect on the bacteria probably due to the formation of supramolecular nanocomplexes based on metal-phenolic coordination (Figure S12).⁵²”

The following literature has been added in the revised manuscript:

52 Fan, G. et al. Protection of anaerobic microbes from processing stressors using metal–phenolic networks. *J. Am. Chem. Soc.* (2021).

The following figure was added as **Figure S12** in the revised supplementary materials:

Figure S12. Log reduction of the TA/Fe^{III}-TA complexes against *EcN* calculated from the CFU counting. The chelation of iron ions with TA reduces the antibacterial ability of TA. Variation is represented by the standard deviation of three independent replicates in all graphs.

Q1-16: Details to prepare this antibiotic-resistant probiotics should be included in the experimental section.

RI-16: We have added the detailed experimental protocol in the revised manuscript and supplementary materials.

Q1-17: In figure 1a, as *E. coli* belong to Gram⁻, Lact belong to Gram⁺, it should not be presented probiotic microbes like the way indicated in this paper.

RI-17: The corresponding texts of **Figure 1a** have been amended accordingly.

Q1-18: In Figure 2a, there is no control for bacteria with dye only to ensure the fluorescent signal is truly from nanoarmor.

RI-18: Thanks for the reviewers for pointing out the control experiment. We have added control experiments by labeling the armored probiotics with BSA-Alexa-647 and Hoechst simultaneously to ensure that the observed red fluorescent signal was from the nanoarmor (new **Figure S2**).

The following figure was added as **Figure S2** in the revised supplementary materials:

Figure S2. CLSM images of armored/naïve *EcN*. Nanoarmor was labeled with bovine serum albumin conjugated with Alexa Fluor 647. *EcN* was labeled with Hoechst. The results showed that BSA-Alexa-647 could not label naïve probiotics.

Q1-19: Fig5-g, why the bacterial counts were lower when there was no antibiotics applied?

R1-19: The number of colonies counted in **Figure 5g** was EcC_{tet} for oral administration. Due to the protective mechanism of the native bacterial community in the GI tract, it is difficult to colonize foreign bacteria without the disruption of native bacterial flora in GI (please see references below, *R1-19_Ref 1– Ref 3*). Briefly, healthy gut microbiota provides protection against the colonization of exotic microorganisms by deploying multiple mechanisms. Changes in microbiota composition, and potential subsequent disruption of colonization resistance, can be caused by antibiotics, thereby providing opportunities for exogenous mechanisms to colonize the gut (please see references below, *R1-19_Ref 4*). Therefore, EcC_{tet} colonized more easily in the intestine of rats treated with antibiotics.

Accordingly, we have added the following sentence in Line 297 on page 15 of the revised manuscript to clarify this information:

“Colonization resistance of healthy gut microbiota led to the relatively low number of EcC_{tet} colonies in the group without the administration of levofloxacin.^{20,53,54}”

In addition, the following literature has been added in the revised manuscript:

53 Volllaard, E. J. & H. A. Clasener. Colonization resistance. *Antimicrob. Agents Ch.* **38**, 409-414 (1994).

R1-19_Ref 1: Suez, J. *et al.* Post-antibiotic gut mucosal microbiome reconstitution is impaired by probiotics and improved by autologous FMT. *Cell* **174**, 1406-1423 (2018).

R1-19_Ref 2: Baruch, E. N. *et al.* Fecal microbiota transplant promotes response in immunotherapy-refractory melanoma patients. *Science* **371**, 602-609 (2021).

R1-19_Ref 3: Vollaard, E. J. & H. A. Clasener. Colonization resistance. *Antimicrob. Agents Ch.* **38**, 409-414 (1994).

R1-19_Ref 4: Ducarmon, Q. R. *et al.* Gut microbiota and colonization resistance against bacterial enteric infection. *Microbiol. Mol. Biol. Rev.* **83**, e00007-19 (2019).

Q1-20: The numbers of how many mice were used in each group and the replicates of each in vivo/vitro experiments were missed in all the figures.

R1-20: Thanks for the reminder from the reviewer. We have added this important information in the revised manuscript.

Response to Reviewer #2:

The authors used polyphenol-based armored probiotics to provide protection against antibiotics that are used simultaneously or after antibiotic therapy. In vitro experiments showed protection from antibiotics while naïve unprotected probiotics did not. In rat challenge experiments, researchers also showed the ability of armored probiotics to withstand antibiotic challenge, survive, flourish and prevent antibiotic-induced diarrhea. Though the approach seems interesting, the lack of mechanistic insights of protection prevents it from fully evaluate its broad utility.

We thank the reviewer for commenting positively on our manuscript. We have conducted a series of additional experiments and obtained new insights to further support our discussions about the protection mechanism endowed by the polyphenol-based nanoarmor. A point-by-point response to the critiques, along with a description of relevant revisions, is below:

Q2-1: Polyphenols including tannic acid used in this study are inhibitory to many bacteria. Some probiotic bacteria such as those used in this study may be resistant; however, many gut bacteria are susceptible. What was the effect of armored bacteria on the gut microbiota of rats?

R2-1: We thank the reviewer for the comment. To better address these comments from the reviewer, we provided our responses into two parts:

1) The results in **Figures 4b,c** showed that the naïve and armored probiotics had similar viabilities and growth curves. These results indicated that polyphenol-based nanoarmor has neglected effect on the growth of armored probiotics. Moreover, a recent report also shows that the polyphenol-based supramolecular nanocoating has no cytotoxicity to the bacteria (*J. Am. Chem. Soc.* 2021, 10.1021/jacs.1c09018). This is probably due to the formation of nanostructured networks around the bacteria through metal-coordination rather than soluble molecules of polyphenols.

The following text has been added in Line 235 on Page 12 in the revised manuscript to clarify the neglected effect of nanoarmor on probiotics:

*“Though natural polyphenols generally possess antibacterial capacity, the formation of nanostructured networks around the cells (**Figures S4, S10**) show neglected effect on the bacteria probably due to the formation of supramolecular nanocomplexes based on metal-phenolic coordination (**Figure S12**).⁵²”*

The following literature has been added in the revised manuscript:

*52 Fan, G. et al. Protection of anaerobic microbes from processing stressors using metal–phenolic networks. *J. Am. Chem. Soc.* (2021).*

The following figure was added as **Figure S12** in the revised supplementary materials:

Figure S12. Log reduction of the TA/Fe^{III}-TA complexes against *EcN* calculated from the CFU counting. The chelation of iron ions with TA reduces the antibacterial ability of TA. Variation is represented by the standard deviation of three independent replicates in all graphs.

2) As for the effect of armored bacteria on the gut microbiota of rats, we performed new assays to detect changes of the intestinal microflora in healthy rats (without administration of antibiotics) after daily administration of armored probiotics for 6 days (new **Figure S29**). Metagenomic sequencing for the fecal bacteria of rats showed that the armored bacteria had no significant effect on the composition of the intestinal flora of rats, though the relative abundance of the different strains showed minor changes.

The following text has been added in Line 375 on Page 20 in the revised manuscript to clarify the effect of armored bacteria on the gut microbiota of rats:

“Metagenomic sequencing for the fecal bacteria of rats showed that the armored probiotics had no significant effect on the composition of the intestinal flora of healthy rats, though the relative abundance of the different strains showed minor changes. (Figure S29)”

The following figure was added as **Figure S29** in the revised supplementary materials:

Figure S29. (a) Taxa summary map of different phyla in gut microbiota before/after the

administration of armored probiotics. (b) Heat map of metagenomic sequencing results of different strains of gut microbiota before/after the administration of armored probiotics.

Q2-2: As stated by the authors, armored protection is only good on the parental strain. However, it is surprising to note that the dividing cells also survive and grow in the presence of antibiotics in vitro experiments (Figure 4e and 4f)! Any explanation?

Q2-3: During antibiotic therapy, antibiotics are maintained at a certain dose level in the body. If the daughter cells do not have armored protection, how well do these probiotics survive in the gut in the presence of antibiotics for colonization and persistence to exert health beneficial effects?

R2-2, 2-3: We thank the reviewer for these comments. Given to the connection of **Q2-2** and **Q2-3**, we combined these two questions and responded accordingly. There is a distinct difference between the present polyphenol-based nanoarmor and previously reported cellular physical barriers which only deliver a short-term protection effect when the cell does not divide. Our polyphenol-based nanoarmor can absorb various types of antibiotics around probiotics through multiple interactions of polyphenols (**Figure 3a-c**, new **Figure S11c**), creating a cytoprotective microenvironment for probiotics. This mechanism enables that this cytoprotective microenvironment remains effective even after the probiotics divide and break through this protective shell, and the absorbed deactivated antibiotics will not harm the probiotics.

Thanks to the helpful suggestion from the reviewer, we conducted a more detailed characterization based on cross-sectional TEM images. The new microscopy data showed that the armored EcN could divide and the shell of nanoarmor was shared by the divided bacteria so that the cytoprotective microenvironment can be maintained even after the cell division (new **Figure S14**).

Moreover, we have conducted additional characterizations on the adsorption ability of the nanoarmor for antibiotics. We tested the removal of the six antibiotics by nanoarmor through high performance liquid chromatography (HPLC) (new **Figure S11**). The results showed that the nanoarmor was able to absorb > 90% of the antibiotics with the concentration of MBC in solution. This makes the residual antibiotics in the solution insufficient to kill the probiotics.

The following text has been added in Line 248 on Page 13 in the revised manuscript to clarify the remaining protective effect of nanoarmor after the division of probiotic cells:

“This result indicated that the nanoarmor could not affect the growth of the bacteria in full media. Cross-sectional TEM images indicated that the shell of nanoarmor could be shared by the divided bacteria so that the protective effect can be maintained even after the cell division (Figure S14).”

In Line 207 on Page 10, the following sentences were added to further strengthen the discussion on the adsorption ability of nanoarmor:

“In addition, we measured the antibiotic concentrations in the supernatants of the solutions treated

with Fe^{III} -TA aggregates by high performance liquid chromatography (HPLC). More than 90% of the antibiotics could be adsorbed by the Fe^{III} -TA aggregates, and therefore the remaining antibiotics were not able to completely kill the probiotic bacteria (**Figure S11**). This result indicated that the protection mechanism of nanoarmor to probiotic bacteria was mainly based on the mechanism of antibiotics adsorption, creating a long-term microenvironment with low antibiotics concentration around the probiotic cells.”

The following figure was added as new **Figure S11** in the revised supplementary materials:

Figure S11. (a) Scheme of assays for the antibiotic adsorption ability of nanoarmor. (b) The adsorption effect of nanoarmor on six antibiotics. (c) The nanoarmor owns the ability to absorb antibiotics by the multiple interactions between the TA and antibiotics. Nanoarmor can absorb more than 90% of antibiotics and create a relatively safe microenvironment for probiotics. Variation is represented by the standard deviation of three independent replicates in all graphs.

The following figure has been added as **Figure S14** in the revised supplementary materials:

Figure S14. The cross-sectional TEM image of EcN divides and breaks through the shell of nanoarmor. The nanoarmor could not affect the growth of the bacteria in media, which may be

because the nanoarmor was simply shed off after the division of bacteria.

Q2-4: Due to inconsistent and unpredictable beneficial effects of probiotics, bioengineering strategies have been introduced to assist probiotics to interact with specific host cell receptors to protect against infectious pathogens (for example, see a recent report by Drolia et al. 2020. Nat. Commun. 11, 6344). In such an application, researchers expressed heterologous bacterial proteins on the surface of probiotics. To broaden the utility of nanoarmor technology, authors need to demonstrate that the nanoarmor does not interfere with bacterial surface protein expression and their adhesion and colonization capabilities in vitro and in vivo.

R2-4: We thank the reviewer for these comments. As we stated in the original manuscript, the formation of nanoarmor was transient and constructed from assembled polyphenol-based nanoparticles around the cells. This assembled nanoarmor can be shared and disassembled when the bacteria divide. Therefore, the nanoarmor does not affect the expression of proteins on the surface of bacteria as well as the normal cell proliferation. Please see our detailed response about the mechanism description in R2-2, 2-3. We also observed the transition moment that armored EcN can divide and break through the shell of nanoarmor by cross-sectional TEM (new **Figure S14**).

The following literature 6 has been added in the revised manuscript to better highlight the importance of maintaining bioactive molecules exchange between the probiotics cells and external environment.

“6 Drolia, R. et al. Receptor-targeted engineered probiotics mitigate lethal Listeria infection. Nat. Commun. 11, 1-23 (2020).”

In terms of the effect of nanoarmor on the colonization capabilities of probiotics, as we explained in the original manuscript and additional responses in R2-2, 2-3, the nanoarmor was transient and can be splitted by the divided bacteria so that the inherent colonization ability of probiotics is not affected after the division. Moreover, even before the cell division, the natural polyphenols of nanoarmor process high fouling ability to proteins and mucous layers of the GI tract, which could facilitate the enhanced colonization of armored probiotics. We performed additional experiments to profile the adhesion of armored bacteria to intestinal mucus of rats (new **Figure S16**). The new experimental results showed that the nanoarmor did not affect the adhesion of probiotics to intestinal mucus. Corresponding experimental details have been added in the revised supporting information as a *test of adhesion ability to intestinal mucus*.

The following text and reference have been added in Line 261 on Page 13 in the revised manuscript to clarify the effect of nanoarmor on the colonization capabilities of probiotics:

*“In vitro assays for the adhesion of naïve/armored probiotics to intestinal mucus of rats revealed that the nanoarmor did not affect the adhesion of probiotics to intestinal mucus, which may be due to the naturally inherent mucoadhesive property of TA from the nanoarmor (**Figure S16**).⁵³”*

The following literature has been added in the revised manuscript:

53 Mikyung S. Tannic acid as a degradable mucoadhesive compound. *ACS Biomater. Sci. Eng.* **2**, 687–696 (2016).

The following figure was added as **Figure S16** in the revised supplementary materials with the related experimental results:

Figure S16. (a) The adhesion ability of naïve /armored probiotics to intestinal mucus of rats and (b) corresponding fluorescent photos. The armored probiotics were able to adhere to intestinal mucus. Therefore, nanoarmor had the potential to promote the colonization of armored probiotics in the intestinal tract. Variation is represented by the standard deviation of three independent replicates in all graphs.

Q2-5: Authors reported that EcC_{tet} survived and replicated in the rat intestine in the presence of levofloxacin. Increased survival and growth of EcC_{tet} in the rat gut could be due to their cross-resistance to levofloxacin. It is generally known resistance to one antibiotic can increase bacterial resistance to another. Have the cross-resistance patterns of this strain to levofloxacin been evaluated? Increased resistance of EcC_{tet} may explain the increased survival capacity of the dividing probiotic strain over time to levofloxacin!

R2-5: We thank the reviewer for the insightful comments on the hypothesis of cross-resistance. Considering that naïve EcC_{tet} also colonized slowly in the intestine of AAD rats (**Figure 5b**), we agree that the development of resistance to levofloxacin by EcC_{tet} may have contributed to its colonization. However, there was a significant difference in the ability of naïve EcC_{tet} and armored EcC_{tet} to colonize the rat intestine. On day 7, rats receiving armored EcC_{tet} had 596% more EcC_{tet} in their feces than rats receiving naïve EcC_{tet}. As a core, our study mainly focused on the ability of the armored probiotics to colonize the intestine of AAD rats, specifically with regard to the changes in colonization ability compared to naïve probiotics. Therefore, we believe that the possible effect of increased levofloxacin-resistance of EcC_{tet} over a short experimental period was minimal and did not affect our original conclusions in this manuscript.

Q2-6: Immunological response and health parameters of rats after feeding with armored probiotic has not been fully evaluated to assess any negative effect on health. Inflammatory and immunomodulatory responses need a full evaluation.

R2-6: We thank the reviewer for the concern of biosafety of armored probiotics. We have added additional characterizations to profile the safety of armored probiotics in detail, including blood biochemical parameters and H&E staining. The additional results showed that the expression of pro-inflammatory and anti-inflammatory factors (IL-6, IL-1 β , TNF- α , IL-10) in the serum of all rats were at normal range, and the blood biochemical indexes in the serum of all the rats were in the normal range (new **Figure S26, S27**). There was no significant difference in all indicators between the rats who received armored probiotics and the control rats. Throughout the experiment, no detrimental physiological effects were observed in any of these animals, which exhibited normal tissue morphology, as assessed by histological staining of fixed tissue sections (new **Figure S28**). The results suggested that armored probiotics presented reliable biosafety.

This important discussion has been added in Line 369 on Page 19 in the revised manuscript to clarify the biosafety of the armored probiotics:

*“In the biological toxicity test, rats were orally dosed with up to 20 mg of armored EcC_{tet} daily. The results showed that armored EcC_{tet} did not show significant biological toxicity to the rats (**Figures S26-S28**). The results showed that the expression of pro-inflammatory and anti-inflammatory factors (interleukin-6, interleukin-1 β , tumor necrosis factor- α , and interleukin-10) in the serum of any of the rats was normal. There was no significant difference in six biochemical indicators between the rats who received armored probiotics and the control rats.”*

The following figures were added as **Figure S26-S28** in the revised supplementary materials with the related experimental results:

Figure S26. IL-6, TNF- α , IL-1 β , IL-10 levels in serum of rats with the administration of armored *EcC_{tet}* daily for 6 days. The results showed that the expression of pro-inflammatory and anti-inflammatory factors (IL-6, IL-1 β , TNF- α , IL-10) in the serum of any of the rats was normal. There was no significant difference in all indicators between the rats who received armored probiotics and the control rats. Variation is represented by the standard deviation of three independent replicates in all graphs.

Figure S27. Biochemical analysis of rats with different administration of armored *EcC_{tet}*: (a) aspartate aminotransferase (AST), (b) alanine aminotransferase (ALT), (c) albumin (ALB), (d) glucose (Glu), (e) urea, (f) creatinine (CREA). There was no significant difference in all indicators between the rats who received armored probiotics and the control rats. Variation is represented by the standard deviation of three independent replicates in all graphs.

Figure S28. Representative histological sections obtained from the tissues of each cohort, visualized with H&E stain. No detrimental physiological effects were observed in any of the animals. Scale bars, 200 μm .

Q2-7: Lack of rigor in diarrhea scoring: Only visual fecal consistency score was used however, other parameters as to water loss, Na^+ levels, hemocult test, etc should strengthen the claim.

R2-7: We thank the reviewer for the helpful suggestions. We have added new assays about stool parameters, including water content, Na^+ levels, and hematochezia status (new **Figure S23**). The new experimental results showed that the feces from the group that had not received antibiotics maintained stable water contents and Na^+ levels throughout the experiment. In contrast, animals that took armored EcC_{tet} showed a significant decrease in fecal water contents and Na^+ levels (positive trends towards the healthy control group) during day 4 to day 11. No occult blood was observed for all of the fecal samples throughout the experiment. These results were consistent with the visual fecal consistency score, further strengthening the conclusion that the armored EcC_{tet} provides therapeutic benefits to the AAD animals.

These important results and corresponding discussion have been added in Line 332 on Page 17 in the revised manuscript to further strengthen the conclusion of therapeutic effects of armored probiotics:

“The feces of animals that had not received antibiotics maintained stable water contents and Na^+ levels throughout the experiment (Figure S23). In contrast, animals that took armored EcC_{tet} showed a more significant decrease in fecal water contents and Na^+ levels during day 4 to day 11, when compared with the group treated with naïve EcC_{tet} . No occult blood was observed for all of the fecal samples throughout the experiment. These results were consistent with the visual fecal consistency score, providing additional evidences to support the therapeutic effects of armored EcC_{tet} for the AAD animals.”

The following figure was added as **Figure S23** in the revised supplementary materials:

Figure S23. (a) Fecal water contents of rats measured on days 4, 9, 11. (b) Fecal Na⁺ levels of rats measured on days 4, 9, 11. (c) Fecal hematochezia status of rats measured on days 4, 9, 11. Variation is represented by the standard deviation of three independent replicates in all graphs.

Q2-8: (Figure 2). The authors indicate that armored *E. coli* Nissle (EcN) recovered from the antibiotics could also be grown in fresh LB culture media, whereas no growth was observed for naïve EcN under the same conditions (Figure S7). However, antibiotic treatments are known to induce sublethal stress in bacteria. Did the authors observe any antibiotic-induced sublethal stress or injury on the armored probiotics? Such stress may reduce the survival of probiotics and eventually their beneficial attributes.

R2-8: We thank the reviewer for these insightful comments on the possible formation of sublethal stress. The protective mechanism of the polyphenol-based nanoarmor has been further discussed and clarified in our detailed responses in **R2-3** and **R2-4**. Briefly, nanoarmor is able to absorb antibiotics near the probiotics, thus creating a microenvironment with low antibiotic concentrations for probiotics. The concentration of the residual antibiotics in this microenvironment was not sufficient to cause antibiotic-induced sublethal stress or serious injury to the probiotics.

According to the reviewer's suggestion, we further carried out a cross-sectional TEM characterization to observe the potential antibiotic-induced sublethal stress or injury of naïve/armored probiotics after treating antibiotics (new **Figure S15**). In this experiment, the armored probiotics did not grow and divide due to the minimal medium condition in the simulated intestinal fluid. After 3 hours of treatment, the morphology of naïve probiotics significantly deformed due to the antibiotic injury. On the contrast, the armored probiotics treated with levofloxacin for 12 hours still maintained the normal morphology. This result demonstrated the effective protection of probiotics by nanoarmor from antibiotic-induced sublethal stress or injury.

The following discussion has been added in Line 253 on Page 13 in the revised manuscript to clarify the antibiotic-induced sublethal stress or injury of naïve/armored probiotics:

“The morphological changes of armored and naïve probiotics were profiled by cross-sectional TEM. After the treatment of levofloxacin for 3, 6, or 12 hours, the morphology of the armored probiotics remained intact, while deformed morphology can be observed in the naïve probiotics due to the killing action (Figure S15). The nanoarmors kept intact due to the significantly slow division rate of bacteria in the simulated intestinal fluid without culture medium.”

The following figure has been added as **Figure S15** in the revised supplementary materials:

Figure S15. The cross-sectional TEM images of naïve/armored probiotics after treatment of simulated intestinal fluid with levofloxacin for different times. The nanoarmors kept intact due to the significantly slow division rate of bacteria in the simulated intestinal fluid without culture medium. Scale bar, 200 nm.

Q2-9: Page 14: “We then examined the protection ability of the armored antibiotic from the administration of antibiotics in vivo.” How did you coat the antibiotic with polyphenol?

R2-9: We thank the reviewer for bring up this typo. The “armored antibiotic” in this sentence should be revised as “armored probiotics”. We have corrected this mistake and checked throughout the revised manuscript carefully.

Q2-10: Fig 5g. Do the bacterial counts represent total counts from luminal content and tissues or tissues alone? If it is from tissues, were the tissues washed before analysis to determine the levels of colonized probiotics counts? Were those probiotics verified to be the parental strain that was introduced, not the resident probiotic strains that have acquired resistance?

R2-10: We thank the reviewer for these insightful comments. The bacterial counts represent the total counts from tissues. Specifically, intestinal tissues were gently washed with PBS (pH 7.4) to clean the contents before homogenization. We have added the detailed description in the revised supporting information.

Moreover, the concentration of tetracycline in tetracycline-selective plates was higher than the MIC concentration so that it is difficult for bacteria to resist such high concentrations, even if they naturally develop resistance to tetracycline through cross-resistance mechanisms in a short period of time. We performed new experiments to exam whether bacteria that can grow on tetracycline-selective plates appear in the feces of rats after continuous administration of levofloxacin (new **Figure S17**). Our new results showed that no colonies could grow on tetracycline-selective plates throughout the experiments. This result indicated that the rat intestinal flora could not develop

resistance to growing on tetracycline-selective plates after the treatment of levofloxacin during the experiments.

This important control experiment and corresponding discussion have been added in Line 307 on Page 16 in the revised manuscript to clarify that the intestinal flora of rats with the stimulation of levofloxacin could not grow on tetracycline-selective plates:

“If the rats received levofloxacin without the administration of EcC_{tet}, no colonies can be observed on the tetracycline-selective plates throughout the experiments (Figure S17).”

The following text has been amended in Line 444 on Page 23 in the revised manuscript:

“After 11 days, the rats were sacrificed and their gastrointestinal tracts were harvested, gently washed with PBS (pH 7.4) to clean the contents, homogenized, and subjected to CFU counting on selective plates to determine the spatial distribution of the bacteria within the gut.”

The following figure has been added as **Figure S17** in the revised supplementary materials:

Figure S17. Fecal CFU counts of bacteria in rats which could grow on the tetracycline-selective plates. If the rats received levofloxacin without the administration of EcC_{tet}, no colonies can be observed on the tetracycline-selective plates throughout the experiments. Variation is represented by the standard deviation of three independent replicates in all graphs.

Q2-11: Page 6 + 9: “The XPS results also supported the presence of polyphenol-based nanoarmor around the bacteria (Figure S5).” What is Fe^{III}-TA? Define at its first appearance!

R2-11: We thank the reviewer for pointing out the confusion of this experimental section. We have added the relevant description in the revised manuscript. The following text has been changed in Line 137 on Page 6 in the revised manuscript:

“The XPS results also supported the presence of ferric ion-tannic acid (Fe^{III}-TA) nanoarmor around the bacteria (Figure S5).”

Q2-12: Method: The detail of the nanoarmor preparation is scanty to be reproduced by others!

R2-12: We have rewritten the preparation section in the revised manuscript and supporting information thoroughly to ensure the details for reproducibility.

Q2-13: Page 18: “The metabolic activity of the armored bacterial cells was measured by CFU counts” – How do you measure metabolic activity by CFU? A misleading statement. Viable but nonculturable cells (VBNC) are metabolically active but do not grow on agar plates without proper activation.

R2-13: Thanks for the correction suggestion. We have changed the relevant description in the revised manuscript.

The following text has changed added in Line 407 on Page 21 in the revised manuscript:

“The viability of the armored bacterial cells was measured by CFU counts.”

Q2-14: Line numbering of the text would be very helpful during the review process.

R2-14: We have added line numbering in the revised manuscript and supplementary materials.

Response to Reviewer #3:

This paper is a very well written paper and outlines a novel approach to live encapsulated bacteria to address the negative impacts of antibiotic induced depletion of gut microbiota. Overall there is extensive experimental data to support the major findings.

Response: We thank the reviewer for these positive comments.

Q3-1: The title of the paper doesn't adequately capture the main observation of the manuscript. I would suggest a change to the title to emphasize the major findings of the study and its potential role as a therapeutic strategy. The term safe and transient are also vague, lacking scientific objectivity and are not demonstrated in a mechanistic sense in this study.

R3-1: We thank the reviewer for these comments. We have removed the inappropriate words of “safe” and “transient” and amended the title in the revised manuscript as below:

“A single-cell nanocoating for antibiotic-resistant probiotics in the gut”

Q3-2: The paper repeatedly refers to the ‘inherently transient’ nature of the coating. I believe the authors should provide outline a more mechanistic basis of the time frame involved in dissolution of the coating in the intestine – this is important to understand the impact clinically. Apparently the coating protects the probiotics but also apparently doesn't reduce its ability to replicate so I feel a discussion around the duration/kinetics of the coating would be helpful.

R3-2: We thank the reviewer for these insightful comments. Yes, unlike previously reported physical encapsulation methods, our nanoarmor is constructed from the assembly of polyphenol-based nanoparticles. The connection between nanoparticles is relatively loose, with a ~ 10 nm gap approximately (new **Figure S10**). This feature makes the nanoarmor relatively easy for the bacteria to break through when they divide. **Figure 4b, c** showed that both armored and naïve EcN exhibited similar viability in the absence of antibiotics. According to the suggestion from the reviewer, we performed additional experiments to profile the microstructural changes of the nanoarmor by using cross-sectional TEM in the new **Figure S14**. Reviewers #1 and #2 have related questions and please see our additional detailed responses in **R1-2, R1-6, R1-7, R1-10 and R2-3, R2-4**.

Specifically, the following sentences have been added in Line 248 on Page 13 and in Line 181 on Page 9 in the revised manuscript to further strengthen the discussion:

*“This result indicated that the nanoarmor could not affect the growth of the bacteria in full media. Cross-sectional TEM images indicated that the shell of nanoarmor could be shared by the divided bacteria so that the protective effect can be maintained even after the cell division (**Figure S14**).”*

“The Brunauer, Emmett and Teller (BET) method of adsorption of nitrogen gas and cross-sectional TEM showed a typical microporous structure of armored probiotics, with a pore diameter ranging

from 2.34 – 10.86 nm (**Figure S10**). These pore sizes are large enough for 200 kDa molecules to pass through⁴³.”

The following figure has been added as **Figure S14** and **Figure S10** in the revised supplementary materials:

Figure S14. The cross-sectional TEM image of EcN divides and breaks through the shell of nanoarmor. The nanoarmor could not affect the growth of the bacteria in media, which may be because the nanoarmor was simply shed off after the division of bacteria.

Figure S10. (a) BET result of armored probiotics, which showed a typical microporous structure with a pore diameter ranging from 2.34 – 10.86 nm. (b) The cross-sectional TEM showed that the nanoarmor was constructed by the assembled nanoparticles which were densely distributed on the cell surface with gaps of up to 10 nm.

Q3-3: A key observation of the study is that the therapeutic benefits of the encapsulated probiotics are only evident in a diseased condition. I am not a fan of the term ‘probiotics’ as it is widely misinterpreted that probiotics are health promoting. This study confirms that the encapsulated bacteria provide a therapeutic benefit to address an iatrogenic disease condition but in fact also demonstrates that administration in a healthy condition does not influence confer improvements in gut microbial composition. The fact that the treatment is only beneficial in the disease condition needs to be very clear throughout. For example the claim that ‘armored probiotics have shown the

ability to maintain a steady state concentration inside the gastrointestinal tract' is not factually correct. What is steady state concentration? Is there a defined steady state level of microbiota? This study demonstrated it speeds up the recovery due to antibiotic treatment but it did not maintain equivalent levels to healthy controls. The authors need to openly acknowledge that such probiotics do not confer any particular health benefits - and are only for a specific disease condition.

R3-3: We thank the reviewer for these comments. We have carefully reworded and specified our description about the benefits of armored probiotics in the revised manuscript. Namely, we emphasized in the conclusion of the original manuscript that armored bacteria only have potential therapeutic effect in the specific iatrogenic circumstance requiring the use of antibiotics and bacteria simultaneously. The following texts have been reworded in the revised manuscript:

“Armored probiotics have shown the ability to colonize inside the gastrointestinal tracts of levofloxacin-treated rats,”

“The CFU counts reached the peak number of 12.88×10^6 (armored) CUF/g of feces on the day 7, and maintained stability on day 8 to day 11, indicating that the nanoarmor shells provided sufficient protection to the EcCtet and successfully colonized in the gut for animals receiving continuous antibiotics.”

“The protection offered by the nanoarmor could persist in vivo after oral administration of enteric capsules loaded with armored bacteria in order to promote the colonization of probiotics in the AAD mammalian GI tract.”

“Despite such armored probiotics do not confer particular health benefits for normal rats, we anticipate that this strategy can be implemented to enhance the efficacy of probiotic treatment regimens where antibiotics must be administered concurrently or in close proximity to the probiotic, such as in the treatment of inflammatory bowel disease.”

Q3-4: Other comments include avoiding non scientific terms such as: 1. notoriously imprecise would be better to be nonspecific in their nature. 2. claims that dysbiosis contribute to diabetes need to be supported with scientific studies. To my knowledge while there is a link between altered my microbiome and disease conditions the suggestion that they contribute to this needs to be supported with scientific evidence. 3. also the argument that probiotics replenish their microbiome this is maybe not an accurate interpretation of the scientific literature. Probiotics are there to improve diversity or restore imbalances.

R3-4: We thank the reviewer for pointing out the inappropriate wording. We have changed the description of “notoriously imprecise” as “nonspecific”, and deleted the relevant description about dysbiosis and diabetes. We have also changed the description of “replenish” to “restore imbalances”.

Q3-5: Methods: the study builds on a previous study referenced in 37 and 38 but I believe the methods and experimental detail on the coating procedures to encapsulate in the nano armor should

be detailed in this manuscript. While the schematic is useful it lacks methodological detail. This should be provided to provide the reader with the opportunity to replicate this procedure.

R3-5: We have rewritten the preparation section in the revised manuscript and supporting information thoroughly to ensure the details for reproducibility.

Q3-6: This study demonstrates that in figure 5G treatment produces levels of cfu in the intense time which are higher than the healthy scenario. The authors claim that their approach is ‘safe’ but have they discussed the potential risks where potentially excessive colonization of the intestine occurs in these treatment groups.

R3-6: The number of colonies counted in **Figure 5g** was EcC_{tet} for oral administration. Due to the protective mechanism of the native bacterial community in the GI tract, it is difficult to colonize foreign bacteria without the disruption of native bacterial flora in GI (please see references below, *R1-19_Ref 1– Ref 3*). Briefly, healthy gut microbiota provides protection against the colonization of exotic microorganisms by deploying multiple mechanisms. Changes in microbiota composition, and potential subsequent disruption of colonization resistance, can be caused by antibiotics, thereby providing opportunities for exogenous mechanisms to colonize the gut (please see references below, *R1-19_Ref 4*). Therefore, EcC_{tet} colonized more easily in the intestine of rats treated with antibiotics and excessive colonization was not observed in our study.

Accordingly, we have added the following sentence in Line 297 on Page 15 of the revised manuscript to clarify this information:

“Colonization resistance of healthy gut microbiota led to the relatively low number of EcC_{tet} colonies in the group without the administration of levofloxacin^{20,54,55}. ”

In addition, the following literature has been added in the revised manuscript:

54 Vollaard, E. J. & H. A. Clasener. Colonization resistance. Antimicrob. Agents Ch. 38, 409-414 (1994).

R1-19_Ref 1: Suez, J. et al. Post-antibiotic gut mucosal microbiome reconstitution is impaired by probiotics and improved by autologous FMT. Cell 174, 1406-1423 (2018).

R1-19_Ref 2: Baruch, E. N. et al. Fecal microbiota transplant promotes response in immunotherapy-refractory melanoma patients. Science 371, 602-609 (2021).

R1-19_Ref 3: Vollaard, E. J. & H. A. Clasener. Colonization resistance. Antimicrob. Agents Ch. 38, 409-414 (1994).

R1-19_Ref 4: Ducarmon, Q. R. et al. Gut microbiota and colonization resistance against bacterial enteric infection. Microbiol. Mol. Biol. Rev. 83, e00007-19 (2019).

REVIEWERS' COMMENTS

Reviewer #1 (Remarks to the Author):

A major concern remains with respect to the rationality of using a polyphenol coating to protect probiotics from antibiotics. Simultaneous administration of probiotics and antibiotics is rarely applied in clinical. Even though, taking turns to administer probiotics and antibiotics is a simple and practical way to avoid the negative effect of antibiotics on probiotics. On the other side, the absorption of antibiotics by polyphenol coating is the key point of this work, as claimed by the authors. However, the elimination of antibiotics by this coating can inevitably reduce the uptake of antibiotics by the gut and diminish the therapeutic efficacy of antibiotics, which raises questions for the rationalities of the coating and the combination of probiotics and antibiotics. Furthermore, the authors have missed important recent advances in the utilization of polyphenol coating (even tannic acid, the same one used in this work) to protect probiotics as well as for oral delivery of therapeutic bacteria.

Reviewer #2 (Remarks to the Author):

NCOMMS-21-23322A

The authors have added additional experimental data and explanations to respond to the comments and made significant improvements in the quality, depth, and readability. The manuscript should be given consideration for publication. However, there are minor issues that need the authors' attention before acceptance.

Line 79: "...to restore imbalances of their microbiome..." or it should read...restore balances of their microbiome!

Line 129: gram-positive – capitalize G – it's a proper name

Line 325: "...tight junction proteins (Occludin, Claudin-1)". Occludin and claudin-1. Use lowercase "o" and "c"

Line 440: who is the vendor for rats?

Line 440-465: How did you gavage capsules? What are the dimensions of capsules and how many were fed per animal?

Reviewer #3 (Remarks to the Author):

The authors have responded appropriately to the comments I made in my first review. I have no further comments.

Bold text indicates reviewer critique

Blue text indicates responses or manuscript revisions. All new inclusions in the revised manuscript are highlighted in *red*.

Our responses to the specific requests made by the three reviewers are as follows.

Response to Reviewer 1:

Q1-1: A major concern remains with respect to the rationality of using a polyphenol coating to protect probiotics from antibiotics. Simultaneous administration of probiotics and antibiotics is rarely applied in clinical. Even though, taking turns to administer probiotics and antibiotics is a simple and practical way to avoid the negative effect of antibiotics on probiotics.

RI-1: We thank the reviewer for the comment. Widespread antibiotic exposure is associated with the emergence of resistant strains and with a variety of gastrointestinal (GI) effects, hypersensitivity, and drug-specific adverse effects, most notably antibiotic-associated diarrhea (AAD) in 5% to 35% of treated humans. Probiotics have been proposed to constitute an effective preventive treatment for antibiotics-induced dysbiosis and associated adverse effects in mice and some human studies (Please see references below, *RI-1_Ref 1*). We believe the strategy of armored probiotics will be a potential way to avoid the negative effect of antibiotics in the intestinal tract.

RI-1_Ref 1: Suez, J. et al. Post-antibiotic gut mucosal microbiome reconstitution is impaired by probiotics and improved by autologous FMT. Cell 174, 1406-1423 (2018).

Q1-2: On the other side, the absorption of antibiotics by polyphenol coating is the key point of this work, as claimed by the authors. However, the elimination of antibiotics by this coating can inevitably reduce the uptake of antibiotics by the gut and diminish the therapeutic efficacy of antibiotics, which raises questions for the rationalities of the coating and the combination of probiotics and antibiotics.

RI-2: We thank the reviewer for bringing up this discussion. We agree that the nanoarmor will absorb a very small amount of antibiotics. However, we believe it does not significantly affect the therapeutic efficacy of antibiotics if we keep the total amount of nanoarmor at a rational level. In fact, this research focused on the colonization and survival of probiotics in the microenvironment with low antibiotic concentration created by the nanoarmor. In future studies, we will pay close attention to the potential effect of nanoarmor on the therapeutic efficacy of antibiotics.

Q1-3: Furthermore, the authors have missed important recent advances in the utilization of polyphenol coating (even tannic acid, the same one used in this work) to protect probiotics as well as for oral delivery of therapeutic bacteria.

R1-3: We thank the reviewer for the suggestion. The following literatures have been added in the revised manuscript to better clarify the recent advances of the bacteria coating for delivery of therapeutic bacteria.

38 Liu J., et al. *Biomaterials coating for on-demand bacteria delivery: Selective release, adhesion, and detachment. Nano Today* **41**, 101291 (2021).

39 Lin, S., et al. *Mucosal immunity-mediated modulation of the gut microbiome by oral delivery of probiotics into Peyer's patches. Sci. Adv.* **7**, eabf0677 (2021).

40 Tang, Y., et al. *Engineered *bdellovibrio bacteriovorus*: a countermeasure for biofilm-induced periodontitis. Mater. Today* (2022). doi.org/10.1016/j.mattod.2022.01.013

49 Fan, G. et al. *Protection of anaerobic microbes from processing stressors using metal-phenolic networks. J. Am. Chem. Soc.* **144**, 2438–2443 (2021).

Response to Reviewer 2:

The authors have added additional experimental data and explanations to respond to the comments and made significant improvements in the quality, depth, and readability. The manuscript should be given consideration for publication. However, there are minor issues that need the authors' attention before acceptance.

Response: We thank the reviewer for these positive comments.

Q2-1: Line 79: "...to restore imbalances of their microbiome..." or it should read...restore balances of their microbiome!

Line 129: gram-positive – capitalize G – it's a proper name

Line 325: "...tight junction proteins (Occludin, Claudin-1)". Occludin and claudin-1. Use lowercase "o" and "c"

R2-1: We thank the reviewer for bringing up these typos. We have corrected these mistakes and checked throughout the revised manuscript carefully.

Q2-2: Line 440: who is the vendor for rats?

R2-2: We thank the reviewer for pointing out the confusion of the experimental details. We have added the related information in the revised manuscript to ensure the details for reproducibility.

The following text has been added in Line 370 on Page 16, and in Line 587 on Page 25 in the revised manuscript:

"Male 8-week-old Wistar rats were purchased from Dashuo Laboratory Animal Technology, Ltd. (China)."

Q2-3: Line 440-465: How did you gavage capsules? What are the dimensions of capsules and how many were fed per animal?

R2-3: We thank the reviewer for these comments. We administrated the capsules by a special dosing tube (Torpac inc, USA). The dimension of the capsules was shown in Line 385 on Page 16 in the revised manuscript: “size type 9h, TORPAC inc, USA”. The parameter of the capsules (size type 9h) could be found in “<https://www.torpac.com/Reference/rat/Rodent%20Catalog.pdf>”. 10 mg of the EcC_{tet} that had been previously armored were lyophilized and filled into Eudragit L100 coated enteric capsule. The dosage of EcC_{tet} was 10 mg (one capsule) per animal daily.

The following text has been added in Line 592 on Page 25 in the revised manuscript:

“For administration, 10 mg of the EcC_{tet} that had been previously armored were lyophilized and filled into Eudragit L100 coated enteric capsules (size type 9h, TORPAC inc, USA).”

The following text has been added in Line 594 on Page 25 in the revised manuscript:

“The enteric capsules containing bacteria were administered daily for each rat by oral gavage for 6 days, during which time the drinking water contained levofloxacin.”

Response to Reviewer 3:

The authors have responded appropriately to the comments I made in my first review. I have no further comments.

Response: We thank the reviewer for the positive comment.